# Trading Place for Space: Increasing Location Resolution Reduces Contextual Capacity in Hippocampal Codes

**Spencer Rooke**[1]    **Zhaoze Wang**[2]    **Ronald W. Di Tullio**[3*]    **Vijay Balasubramanian**[1,4,5*]

[1]Departments of Physics    [2] of Computer and Information Science and [3] Neuroscience
University of Pennsylvania; [4]Rudolf Peierls Centre for Theoretical Physics, University of Oxford;
and [5]Santa Fe Institute    *: Equal contribution
srooke@sas.upenn.edu    zhaoze@seas.upenn.edu
ron.w.ditullio@gmail.com    vijay@physics.upenn.edu

## Abstract

Many animals learn cognitive maps of their environment - a simultaneous representation of context, experience, and position. Place cells in the hippocampus, named for their explicit encoding of position, are believed to be a neural substrate of these maps, with place cell "remapping" explaining how this system can represent different contexts. Briefly, place cells alter their firing properties, or "remap", in response to changes in experiential or sensory cues. Substantial sensory changes, produced, e.g., by moving between environments, cause large subpopulations of place cells to change their tuning entirely. While many studies have looked at the physiological basis of remapping, we lack explicit calculations of how the contextual capacity of the place cell system changes as a function of place field firing properties. Here, we propose a geometric approach to understanding population level activity of place cells. Using known firing field statistics, we investigate how changes to place cell firing properties affect the distances between representations of different environments within firing rate space. Using this approach, we find that the number of contexts storable by the hippocampus grows exponentially with the number of place cells, and calculate this exponent for environments of different sizes. We identify a fundamental trade-off between high resolution encoding of position and the number of storable contexts. This trade-off is tuned by place cell width, which might explain the change in firing field scale along the dorsal-ventral axis of the hippocampus. We demonstrate that clustering of place cells near likely points of confusion, such as boundaries, increases the contextual capacity of the place system within our framework and conclude by discussing how our geometric approach could be extended to include other cell types and abstract spaces.

## 1 Introduction

Decades of experiments suggest that the mammalian hippocampus is crucial for the formation of episodic memories and spatial navigation [1, 2, 3]. Neural recordings of rodents during active navigation led to the discovery of place cells by John O'keefe [4], named for their spatially localized firing patterns. These place cells were quickly theorized to be the substrate of the cognitive map - an animal's simultaneous and abstract representation of context, experience, and position [3, 5]. Further experiments led to the discovery of remapping [6, 7, 8], during which place cells alter their firing properties in response to changes in sensory and contextual cues. Large contextual changes lead to global remapping, in which population level maps of activity appearing in different contexts are nearly orthogonal, independent of correlations within an environment [9, 10]. Many have speculated that

38th Conference on Neural Information Processing Systems (NeurIPS 2024).

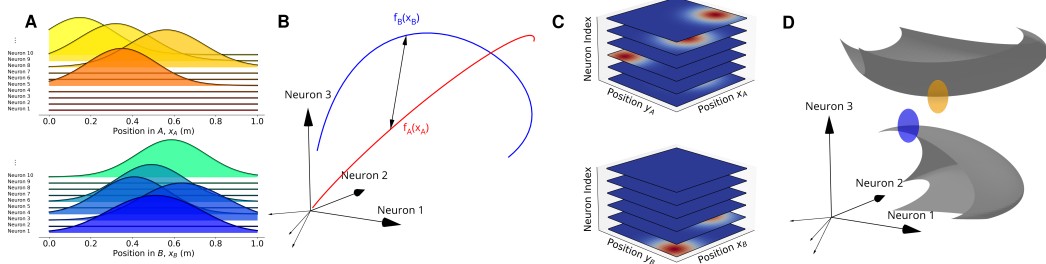

Figure 1: Place cell firing fields remap in one **(A)** and two **(C)** dimensions. **(B)** The maps $f_A$, $f_B$ of one dimensional contexts correspond to curves in neural population activity space, parametrized by position. With constant Gaussian noise, we require that these curves be a distance $2\sigma(\sqrt{N} + q)$ apart in order to discriminate contexts. **(D)** The maps $f_A$, $f_B$ of two dimensional contexts correspond to surfaces in activity space, parametrized by position in physical space. With activity dependent, Poisson-like noise, firing patterns exponentially localize to characteristic ellipsoids. We require that these thickened surfaces do not intersect in order to discriminate contexts.

the place cell system encodes context via global remapping and that this encoding scheme should be able to store a large number of contexts [11, 12], but precise calculations backing these speculations are lacking. Here, we analyze the geometry of hippocampal codes to approximate the capacity and properties of context encoding by place cells.

We treat place cell population activity as a high dimensional space into which we can embed contexts (**Fig. 1 ,B**). Since a large fraction of hippocampal neurons are place cells [13, 14], we expect this activity space to have thousands of dimensions, depending on the species. While the defining features that distinguish context are not fully known [6, 7, 8], we define it as a choice of surrounding environment, as is often done in practice in experiments [9, 11, 15, 16, 17, 18, 19]. Embedding an environment in the firing rate space produces a neural manifold, with position on the manifold corresponding to particular patterns of neural population activity that reflect location in real space. Without noise, the dimensionality of this manifold matches the dimensions of the encoded environment. For example, different linear tracks would be encoded as different 1D curves embedded in the population activity space (**Fig. 1B**). In this geometric framework, the distance between two manifolds indicates how differently the neural populations encode each environment [20] (**Fig. 1B**); and if two manifolds do not overlap at any point, the underlying context can always be determined without confusion. In the noiseless case, we expect that it is trivial to discriminate context because pairs of very low dimensional manifolds are unlikely to overlap in such a high dimensional space. Real neurons, however, will have noisy responses which "blur" our manifolds, giving them a characteristic width around the noiseless, low dimensional embedding of an environment. As such, our criterion for separability of contexts requires that these thickened manifolds for each context do not overlap extensively in rate space (**Fig. 1D**). We consider two models of neuronal noise for our rate based neurons: one in which noise is additive and constant, and a second in which noise is additive, but with variance scaling with neural activity. The latter mirrors an underlying Poisson-like process of spike generation that is often assumed for hippocampal neurons. We investigate the effect of these different noise models on pairwise overlap, and from this extrapolate the probability that multiple contexts are discriminable.

When the number of neurons $N$ is large, these manifolds grow far apart, and are more easily distinguished. In weak to moderate noise regimes, we find that the number of contexts that can be stored by a place-like coding scheme grows exponentially in $N$, even when we enforce strict requirements that the system can discriminate between contexts at *any* position within an environment. We further find that place cell width tunes both the ability to decode local position and the typical distance between contextual manifolds. Large place cell widths generate greater overlap between firing fields, which in turn increases discriminability between contexts. Conversely, small place cell widths increase the spatial resolution of encodings, but decrease the separability of contextual manifolds. This leads to a fundamental trade-off between decoding context and local position. We propose that this trade-off accounts for the observed change in firing field width across the dorsal-ventral axis in the rodent hippocampus, and predict that selective inhibition along the hippocampus will lead to

different types of memory impairment for spatial tasks, consistent with existing experimental evidence [21, 22, 23, 24, 25, 26]. Finally, we find that when fields are uniformly distributed, confusion between contexts is most likely to occur near boundaries. This effect can be compensated for by biasing the density of place field centers towards the boundary, recreating a known feature of rodent place coding [27, 28]. We predict that the observed place cell clustering near boundaries allows the place system to segregate different contexts with greater efficiency, and the extent of this clustering is additionally a function of cell location along the ventral-dorsal axis.

## 1.1 Model Description

We consider a population of $N$ place cells, indexed by $j$, with activity described by a population activity vector $\vec{r}$. We treat this vector as a random variable that depends both on context ($A$), and position within a context ($x_A$). We consider physical environments in one or two dimensions, as this is common experimentally, so that $x_A$ is either a one or two dimensional vector describing location. Each context $A$ is equipped with a place map $f_A$, which defines the tuning of each neuron when the animal is within $A$. That is, $f_A$ sets the mean firing rate of each neuron, $\langle r_j(x_A, A) \rangle = f_{A,j}(x_A)$. Each map $f_A$ can be viewed as an encoding for a particular context that embeds $x_A$ into a certain set of population activity vectors $\vec{r}$ (**Fig. 1A,B**). We restrict our analysis to rate coding models of population activity, where $\vec{r}$ represents population firing rates (rather than spike counts) and has additive noise.

We assume that place maps are constructed stochastically for each environment, consistently with known properties of place cells. Within an environment, each place cell has $a_j$ distinct firing fields, where $a_j$ is drawn from a gamma-poisson distribution, following recent experimental observations in rodents [29, 30], (**Supplemental**). For small environments (1-2 m), typical place cells have 0-2 firing fields under these statistics, with greater recruitment as environment size increases. We give each firing field a gaussian shape, and vary the widths parametrically. The tuning curve of each neuron is then a sum of gaussians:

$$f_{A,j}(x_A) = .1Hz + C_{j,A} \sum_{i}^{a_j} \exp -\frac{1}{2(w_i/2)^2}(\vec{x}_A - \vec{\mu}_{A,i})^2 \tag{1}$$

For simplicity, the normalization $C_{j,A}$ is chosen so that all neurons have a baseline firing rate of $.1Hz$, and a maximum firing rate of $30Hz$ in environments in which they are active.

With additive noise, neural activity is given by $r_j(x_A, A) = f_{A,j}(x_A) + \xi_j$. We consider two noise model. In the first, $\xi$ is gaussian distributed, and noise is not correlated between place cells, so that $\vec{\xi} \sim \mathcal{N}(0, \sigma^2 \mathbb{I})$. Note that $\sigma^2$ is the variance in noise magnitude and thus has units of $Hz^2$. In the second model, we scale the noise variance of each place cell with activity to match underlying Poisson-like statistics associated with spike generation. In this case, $\xi_j \sim \mathcal{N}(0, \phi f_{A,j}(x_A))$. Here, $\phi$ is a dispersion coefficient that sets how noisy the place system is, and has units of $Hz$. We can write our two models of place cell activity as $\vec{r}(x_A, A) \sim \mathcal{N}(f_A(x_A), \sigma^2 \mathbb{I})$ and $\vec{r}(x_A, A) \sim \mathcal{N}(f_A(x_A), \phi \text{diag}[f_A(x_A)])$, respectively. Additive noise can generate negative firing rates, and so we rectify all negative firing rates to 0 Hz.

## 2 Results

### 2.1 Place Coding Can Store Exponentially Many Contexts

With the above model in hand, we sought to explore the context coding capacity of the place cell network. To do so, we first defined what it means to discriminate two contexts $A$ and $B$. In our formulation, $f_A(x_A)$ and $f_B(x_B)$ define two manifolds in the space of neural activity (**Fig. 1**). Without noise, if these two manifolds do not intersect, then the firing patterns that arise in each environment are unique to that environment. In this case, both position and context can be uniquely determined from population activity, and so the contexts $A$ and $B$ are discriminable so long as $f_A$ and $f_B$ do not intersect. For the moment, we disregard considerations of computational complexity required for manifold discrimination, such as requiring linear separability as in [31, 32].

When there is no noise, there is a very low probability that the surfaces defined by $f_A$ and $f_B$ intersect at any point in our high-dimensional firing rate space, and the intersection criterion is trivial to fulfil.

The introduction of noise gives the manifolds defined by $f_A$ and $f_B$ a characteristic width, whose scale and geometry depends on the nature of the noise. In the model where noise variance $\sigma^2$ is constant, the characteristic manifold width scales like $\sigma\sqrt{N}$ when $N$ is sufficiently large. This is due to a well known characteristic of gaussians in high dimensions, in which normal distributions have the majority of their mass sitting near a thin annulus of radius $\sigma\sqrt{N}$ [33, 34] (**Supplemental**). That is, the probability density for the radius of vectors pulled from high dimensional spherical gaussians peaks at $\sigma\sqrt{N}$, and falls off exponentially away from this shell as $p_R(\sigma\sqrt{N}+q\sigma) \approx p_R(\sigma\sqrt{N})e^{-q^2}$. For example, when $q = 2$ this leads an approximate four e-fold decrease in the probability density, so that the majority of the probability mass ($> 99\%$ for large $N$) is within a radius of $\sigma\sqrt{N} + q\sigma$. Importantly, this exponential fall off is independent of $N$, and so the width of the annulus is of order 1 in $N$. In the rate dependent noise model, we instead get a probability mass that is exponentially localized to an ellipsoid with major axes whose lengths depend on firing rates, $\sqrt{N\phi f_A^i(x_A)}$.

We would then like a way to determine the effect of both noise models on manifold overlap, and by extension, decrease in context discriminability. We solve this problem geometrically. For the rate independent (gaussian) noise model, the manifold $f_A(x_A)$ acquires a width of $\sigma(\sqrt{N} + q)$ in every direction. Here, $q$ accounts for the non-zero width of the noise annulus. In any case, our condition that two contexts $A$ and $B$ be distinguishable then becomes a requirement that the minimum distance between $f_A(x_A)$ and $f_B(x_B)$ in rate space overcomes this width set by noise (**Fig. 1C**):

$$\min_{x_A, x_B} d(f_A(x_A), f_B(x_B)) > 2\sigma(\sqrt{N} + q) \tag{2}$$

The manifold width acquired from the rate dependent (Poisson-like) noise model will vary in each direction of the firing rate space with the neural firing rate. Intuitively, we can think of this widening of the manifold as placing an ellipsoid at each point along our manifold, with principle axes set by the firing rates of each neuron (**Fig. 1D**). We then check if our thickened manifolds overlap. Unlike the case with constant noise, where it sufficed to check that the distance between points between different manifolds is greater than the minimum distance set by noise, we must check that the ellipsoids centered at $f_A(x_A)$ and $f_B(x_B)$ do not overlap for any pair $x_A, x_B$ on the two manifolds. For ellipsoids with centers $\vec{\mu}_A = f_A(x_A)$ and $\vec{\mu}_B = f_B(x_B)$ and covariances $\Sigma_A$ and $\Sigma_B$, we can define the set:

$$\mathcal{E}_s = \{s(\vec{r} - \vec{\mu}_A)^T \Sigma_A (\vec{r} - \vec{\mu}_A) + (1 - s)(\vec{r} - \vec{\mu}_B)^T \Sigma_B (\vec{r} - \vec{\mu}_B) \leq 1\} \tag{3}$$

Here, $s \in [0, 1]$. For $s = 0$ or $s = 1$, this set describes the interior of the ellipsoid centered at $\vec{\mu}_A$ or $\vec{\mu}_B$, respectively, and varying $s$ interpolates between the two. For other values of $s$, $\mathcal{E}_s$ is either empty, a single point, or the interior of an ellipse. We also note that, for any $s$, the intersection of the two ellipsoids is always contained in $\mathcal{E}_s$, and so if there is an $s$ for which this set disappears, then the two ellipsoids do not intersect [35]. We can rewrite $\mathcal{E}_s$ as

$$\mathcal{E}_s = \{(\vec{r} - \vec{\mu}_s)^T \Sigma_s (\vec{r} - \vec{\mu}_s) \leq K(s)\} \tag{4}$$

where $\Sigma_s = s\Sigma_A + (1 - s)\Sigma_B$ and $\vec{\mu}_s = \Sigma_s^{-1}(s\Sigma_A\vec{\mu}_a + (1 - s)\Sigma_B\vec{\mu}_b)$. Thus, the two ellipsoids centered at $\vec{\mu}_A$ and $\vec{\mu}_B$ do not intersect if and only if $K(s)$ is negative for some $s \in [0, 1]$; i.e., $\mathcal{E}_s$ is empty for some $s$. As the centers of each ellipsoid are a function of position within their respective contexts, we find that (**Supplemental**):

$$K(s, x_A, x_B) = 1 - \frac{1}{\phi(\sqrt{N} + q)^2} \sum_i^N \frac{(f_B^i(x_B) - f_A^i(x_A))^2}{(\frac{1}{1-s}f_A^i(x_A) + \frac{1}{s}f_B^i(x_B))} \tag{5}$$

If for every pair $x_A, x_B$, there is an $s$ for which this is negative, then our two thickened manifolds do not intersect. As such, we let $s^*(x_A, x_B)$ minimize $K(s, x_A, x_B)$ for each choice of $x_A, x_B$. To put this condition in a similar form as equation (2), we define $\phi^*$:

$$\phi^*(x_A, x_B) = \frac{1}{N} \sum_i^N \frac{(f_B^i(x_B) - f_A^i(x_A))^2}{(\frac{1}{1-s^*(x_A,x_B)}f_A^i(x_A) + \frac{1}{s^*(x_A,x_B)}f_B^i(x_B))} \tag{6}$$

Our condition on $K(s, x_A, x_B)$ can then be written in a similar form to the simpler noise model:

$$\min_{x_A, x_B} N\phi^*(x_A, x_B) > (\sqrt{N} + q)^2\phi \tag{7}$$

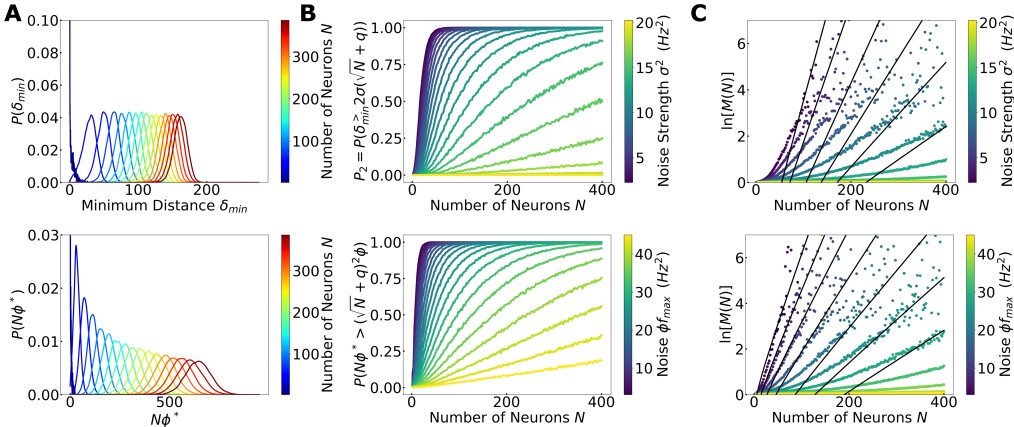

Figure 2: **(A)** The distributions for the minimum distance in rate space of $\delta_{min}$ (Top) and the analogue used for the rate dependent noise model, $N\phi^*_{min}$ (Bottom), constructed from kernel density estimates. At large $N$, these approach gaussian. Plots are for one dimensional rooms, with room length $L = 1m$ and firing field widths $W = 1/3m$. **(B)** The probability that two contexts are distinguishable as a function of the number of neurons and at different noise levels, for the rate independent (Top) and dependent (Bottom) noise models. **(C)** The logarithm of $M(N)$, the number of storable contexts as a function of $N$, at $P_M = .95\%$ confidence. In black is the predicted large $N$ scaling, $\gamma N + \frac{1}{4}\ln N$.

Here, $q$ again accounts for the nonzero width of the noise annulus. For both models, the width of the annulus $q$ is $\mathcal{O}(1)$ in $N$, and so makes no contribution to our final results at large $N$. Indeed, we performed the numerical calculations for multiple values of $q$, and find that at large $N$ our results are unchanged. In generating our figures, we use $q = 2$. Note that we are enforcing a very strict definition of separability for both noise models. When separation between the two manifolds is greater than the threshold distance, the probability of confusing the two contexts is vanishingly small. Less strict conditions would still allow for good (in a practical sense) performance in context discrimination, but our stricter definition engenders several advantages. First, the geometric approach we are using to assess capacity allows us to easily generalize to more than two contexts. Second, it is important to be as conservative as possible in capacity calculations and such strictness should prevent overestimation of the number of decodable contexts. Finally, our approach avoids ceiling effects for changing parameters of the model; that is, by making the task as hard as possible we can observe impacts of changing different parameters on performance that would be hidden by performance plateaus on easier tasks.

Having these pairwise separability conditions for two contexts, we then wanted to determine how many contexts $M$ are storable by the place cell system. To do so, we first replace our statements for particular environments $A$ and $B$ with probabilistic statements for any pair of rooms $A$ and $B$ generated at random. The probability that two rooms are distinguishable is then the probability that the following pairwise separation conditions are true:

Constant Noise: $P(2 \text{ Rooms are Separable}) = P(\min_{x_A, x_B} d(f_A(x_A), f_B(x_B)) > 2\sigma(\sqrt{N} + q))$ (8)

Variable Noise: $P(2 \text{ Rooms are Separable}) = P(\min_{x_A, x_B} N\phi^*(x_A, x_B) > (\sqrt{N} + q)^2\phi)$ (9)

For notational convenience, we define $\delta_{min} \equiv \min_{x_A, x_B} d(f_A(x_A), f_B(x_B))$ and $\phi^*_{min} \equiv \min_{x_A, x_B} \phi^*(x_A, x_B)$. The probability that two rooms are distinguishable can then be written in terms of distributions over these variables as:

$$\text{Constant Noise: } P_2 = 1 - \int_0^{2\sigma(\sqrt{N}+q)} P(\delta_{min})d\delta_{min} \tag{10}$$

$$\text{Variable Noise: } P_2 = 1 - \int_0^{\phi(\sqrt{N}+q)^2} P(N\phi^*_{min})Nd\phi^*_{min} \tag{11}$$

That is, we can determine $P_2$ as long as we can calculate the distributions $P(\delta_{\min})$ and $P(\phi^*_{\min})$. These distributions approach normal distributions for large $N$ (**Fig. 2A**, **Supplemental**). We use

numerical methods to find the mean and variance of these distributions. That is, we generate many pairs of rooms with unique place maps for each room and then reconstruct these underlying distributions using normalized Kernel Density Estimation (KDE) while varying the value of $N$ (the number of neurons). Finally we can use these reconstructed distributions to calculate $P_2$ for both noise models as a function of $N$ and the strength of the noise (**Fig. 2B**).

Given the above, we can estimate the total number of storable contexts, $M$, of the place system as a function of key parameters of the system. First, we determine how $M$ scales with the number of neurons $N$. Given the probability that any pair of rooms is distinguishable, we can estimate the probability that $M$ environments are distinguishable, $P_M$, via a union bound:

$$P(\text{M rooms are distinguishable}) = P(\cup_{i \neq j} \text{ rooms } i \text{ and } j \text{ are distinguishable}) \leq P_2^{\binom{M}{2}} \quad (12)$$

In weak to moderate noise regimes, this inequality becomes approximately saturated (**Supplemental**). Thus, we have $P_M \approx P_2^{\frac{M(M-1)}{2}}$. To find the number of storable contexts given $N$ neurons, $M(N)$, we can increase $M$ until $P_M$ falls below a desired confidence or allowable error, and call the $M$ where this occurs the number of storable contexts. For numerically derived values, we use $P_M = .95$, but note that this only changes prefactors, and the scaling behaviour is independent of this choice. Equivalently, we can simply invert $P_M \approx P_2^{\frac{M(M-1)}{2}}$ to find $M(N)$ for a given confidence $P_M$ (**Supplemental**):

$$M(N) \approx \sqrt{2\frac{\log(P_M)}{\log(P_2(N))} + \frac{1}{4}} + 1/2 \sim (N^{1/4} + \mathcal{O}(N^{1/8}))e^{\gamma N} \quad (13)$$

In the limit of a system dominated by noise, we can never meet our geometric constraints, and $M(N) = 1$. However, if the noise is more reasonable, we find that $M$ scales exponentially with $N$ for both noise models (**Fig. 2C**, **Supplemental**). Here, $\gamma$ is a constant that depends on firing field widths, noise, and room geometry but, critically, is independent of $N$ at large $N$. We can calculate $\gamma$ in terms of the distributions of $\delta_{min}$ and $N\phi^*_{min}$ as (**Supplemental**):

$$\gamma_\delta = \left(\frac{\mathbb{E}[\delta_{min}]/\sqrt{N} - 2\sigma}{2\sqrt{\text{Var}[\delta_{min}]}}\right)^2 \qquad \gamma_\phi = \left(\frac{\mathbb{E}[\phi^*_{min}] - \phi}{2\sqrt{N\text{Var}[\phi^*_{min}]}}\right)^2 \quad (14)$$

Here, $\gamma_\delta$ and $\gamma_\phi$ refer to the exponents in the fixed noise model and variable noise model, respectively. One can readily show that at large $N$, the equations for gamma for both noise models will become independent of $N$ (**Supplemental**). In this large $N$ regime, we can then characterize the number of storable contexts solely using the mean and variance of the distributions $P(\delta_{min})$ and $P(N\phi^*_{min})$. We accordingly calculated the number of distinguishable contexts numerically, and compared with the predicted large $N$ behaviour (**Fig. 2C**), finding good agreement. Our results demonstrate that place coding allows encoding of exponentially many contexts with an exponent controlled by the amount of neuronal noise (**Fig. 3**).

## 2.2 A Trade-off Between Spatial Specificity and Context Segregation

Realistic hippocampal place cells have tuning curves of varying widths. Indeed, across the dorso-ventral axis of the hippocampus, place field widths can vary by nearly an order of magnitude [21, 24], with ventral place cells having wider tuning than dorsal cells. As such, we next explored how the exponent of the number of stored contexts $\gamma$ scales as a function of firing field width (**Fig. 3**, **Supplemental**). To do so, we numerically reconstructed the distributions $P(\delta_{min})$ and $P(N\phi^*_{min})$ for various place cell widths. In our model, increasing the widths of place field tunings starting from small sizes generally leads to an increase in the distance between representations of environments.

That is because, especially in small environments, a relatively small number of place cells will show place fields, and hence small place fields lead to sparse population activity, reducing the absolute distance between firing vectors in different environments. Increasing place field widths increases the average neuronal activity within an environment, thus pushing the encoding manifolds apart. As a result, we expect larger fields to increase contextual capacity. In fact, in small environments ($1m - 4m$), optimal context discrimination performance occurs with firing fields that are about the size of the environment (**Fig. 3**). We can get a sense of why this happens as follows. Consider smaller environments in which less than half of all place cells are active due to the Gamma-Poisson

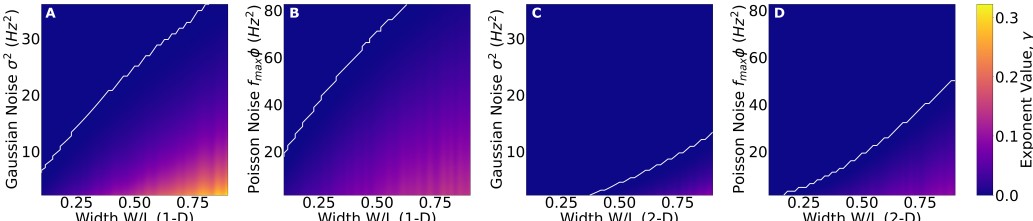

Figure 3: The calculated value of the exponential $\gamma$ at large $N$ of equation (14), as a function of firing field width and neuronal noise. The environments are $1m$ **(A and B)** and $1m^2$ **(C and D)**. **(A and C)** represent the rate independent noise model (Gaussian), while **(B and D)** are the noise dependent model (Poisson-like). White lines demarcate the transition into the non-separable regime. We find better performance for the lower dimensional environments and the Poisson-like model. For larger environments, smaller relative widths become preferable (**Supplemental, Fig. 8**).

statistics of place cell activation [29, 30]. When place cells have extremely large widths in this regime, each neuron is either completely on or off within a given environment, and so each context becomes associated with a random binary identifier, that will be unique with high probability. Thus the environmental context can be read off simply by noting which place cells are active, although there is no location resolution at all. By contrast, in larger rooms, most place cells will have at least one firing field. So, although the sparse firing of narrow place fields makes it harder to discriminate contexts based on their responses, the largest, environment-sized firing fields also become useless in this case because essentially all cells will be active in every room (**Supplemental**). In either case, we expect a tradeoff between the twin goals of context and location discrimination that depends directly on place field size, and indirectly on environment size due to the gamma-poisson statistics used to generate place field centers.

Tuning the firing field widths lets us explore the trade-off between two presumed objectives for hippocampal function; encoding of position and encoding of multiple contexts. While wider fields are generally better for context segregation, it is clear that they are not optimal for spatial specificity, as wider fields result in less variation in population level firing between locations. Thus we expect a trade-off between spatial information encoded within a context, and the ability to separate contexts, tuned by the widths of place cell firing fields. To formalize this trade-off, we must determine how we will explicitly characterize both spatial specificity and context segregation. To characterize spatial specificity, we chose to utilize average decoding performance on decoding current position $\hat{x}$ from the firing rates $\vec{r}(x)$ of the place cell system. Naturally, good performance occurs when the decoded position typically agrees with the true position, or $\langle (\hat{x} - x)^2 \rangle_{\vec{r}|x}$ is small at most positions. To avoid a particular choice of decoder, we invoke the Cramer-Rao bound, which lower bounds the covariance of *any* estimator by the inverse Fisher Information:

$$\langle (\hat{x} - x)(\hat{x} - x)^T \rangle_{P(r|x)} \geq \mathcal{I}^{-1}(x) = (\mathbb{E}_r[(\nabla_x l) \otimes (\nabla_x l)])^{-1} \tag{15}$$

$$\langle (\hat{x} - x)^2 \rangle_{P(r|x)} \geq \mathrm{Tr}[\mathcal{I}^{-1}(x)] \tag{16}$$

When $x$ represents position in a one dimensional context, the inverse Fisher Information is a scalar. If $x$ is not one dimensional, then the inequality is a statement about the difference between the covariance and the inverse of the Fisher Information being positive semi-definite, so that a bound on the mean squared error can be found by taking a trace. In both noise models, the Fisher Information can be calculated exactly in terms of the tunings of each neuron within an environment as (**Supplemental**):

$$\mathcal{I}_\delta = \frac{1}{\sigma^2} \sum_i \nabla_x f_A^i \otimes \nabla_x f_A^i \qquad \mathcal{I}_\phi = \sum_i \left( \frac{1}{\phi f_A^i(x)} + \frac{1}{2 f_A^i(x)^2} \right) \nabla_x f_A^i \otimes \nabla_x f_A^i \tag{17}$$

We can now characterize spatial specificity by calculating the average spatial resolution by averaging the Cramer-Rao bound with respect to both position and context. The objective for maximizing the spatial resolution can be formalized by minimizing $\ln \langle \langle \mathrm{Tr}[\mathcal{I}^{-1}(x)] \rangle_x \rangle_A$ (**Fig. 4A**). Tuning firing fields for high spatial resolution drives the firing field widths to a minimum set by the population size. Clearly, this is at odds with the first objective for storing many contexts, which drives firing fields to be larger. Here we find our anticipated firing field width trade-off between these two objectives. The character of this trade-off depends on how we formalize an objective function with respect to firing

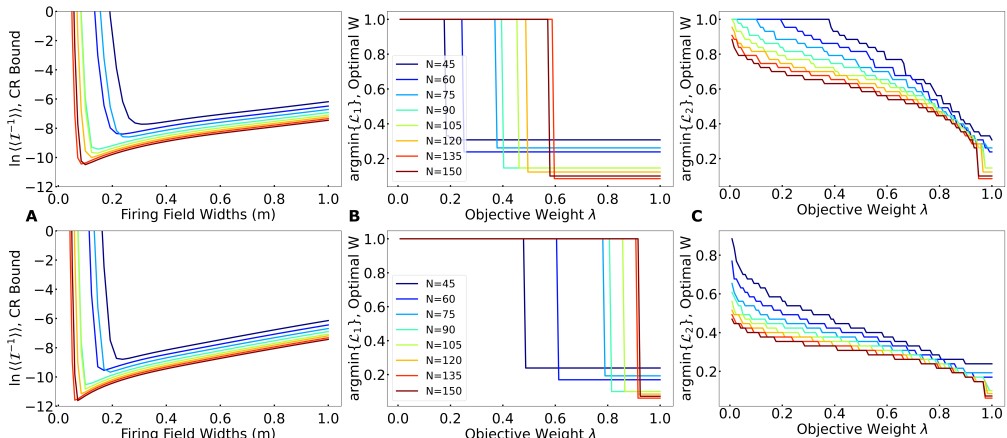

Figure 4: **(A)** The Cramer Rao Bound for both noise models. **(Top)** represents the rate independent model, while **(Bottom)** represents the rate dependent model. **(B-C)** The optimal width as a function of the relative importance between the two objectives. Using $\ln M$ to characterize context decoding leads to a sharp change in the optimal width, while $\ln P_2$ leads to a more gradual change. In both cases, there is a trade-off between storing high resolution information and storing many contexts, tuned by firing field width.

field width $W$. We consider two objective functions:

$$\mathcal{L}_1 = \lambda(-1)\log(M(N,W)) + (1-\lambda)\log\langle\langle\mathrm{Tr}[\mathcal{I}^{-1}]\rangle_x\rangle_A \tag{18}$$

$$\mathcal{L}_2 = \lambda(-1)\log(P_2(N,W)) + (1-\lambda)\log\langle\langle\mathrm{Tr}[\mathcal{I}^{-1}]\rangle_x\rangle_A \tag{19}$$

Here $\lambda \in [0,1]$ interpolates between the two objectives by setting the relative importance of each, $N$ is the number of neurons, $W$ is the width of the firing fields, $P_2$ is the probability that 2 rooms are separable given $N$ and $W$, and $M(N,W)$ is the number of storable contexts (see discussion below eq. 12). As the relative importance shifts from a high contextual capacity to high contextual resolution, the optimal firing field width shrinks to a minimum set by the averaged Cramer-Rao bound. Such capacity-resolution trade-offs are consistent with those demonstrated in recurrent neural networks [36]. For the first choice of objective function, the optimal width jumps abruptly as we vary $\lambda$ (**Fig. 4B**). The second choice of objective function, on the other hand, strongly penalizes widths for which context segregation becomes impossible, leading to a smooth transition of the optimal firing field as we vary $\lambda$ (**Fig. 4C**). Regardless, we see the same clear trade-off between our two objectives for each choice of formalization. This result suggests that the difference in field size across the dorsal-ventral axis of the hippocampus may reflect a segregation of coding function by optimizing for different objectives rather than just a gradient of spatial resolution as is commonly posited [17, 11]. This is also consistent with experimental evidence that dorsal hippocampus is largely recruited for spatial tasks, while ventral hippocampus typically shapes contextual response [22, 23, 24, 25, 26].

## 2.3 Field Clustering near Boundaries improves Context Segregation

So far, we have assumed that the firing field centers are uniformly distributed. In reality, place cells often drift near positions of interest and frequented locations, such as boundaries or rewards. If place cells form a compressed representation of experience, then we can reasonably propose that the density of place cells at a location should reflect an increased resolution for memory formation near that location. This clustering could also have an effect on context separability. Indeed, we predict that such clustering improves context segregation, and demonstrate that biasing place cells towards the boundaries of contexts can improve the ability of the place system to discriminate between them.

A uniformly distributed place cell population will, in general, have less overlapping fields near the boundaries of an environment. Since discrimination between contexts critically depends on the overlap between place cell firing fields, this lack of density by the boundaries increases the probability of confusion between contexts. This observation matches perceptual intuition. In the center of environments, animals are able to reference distal cues as well as different proximal cues in the

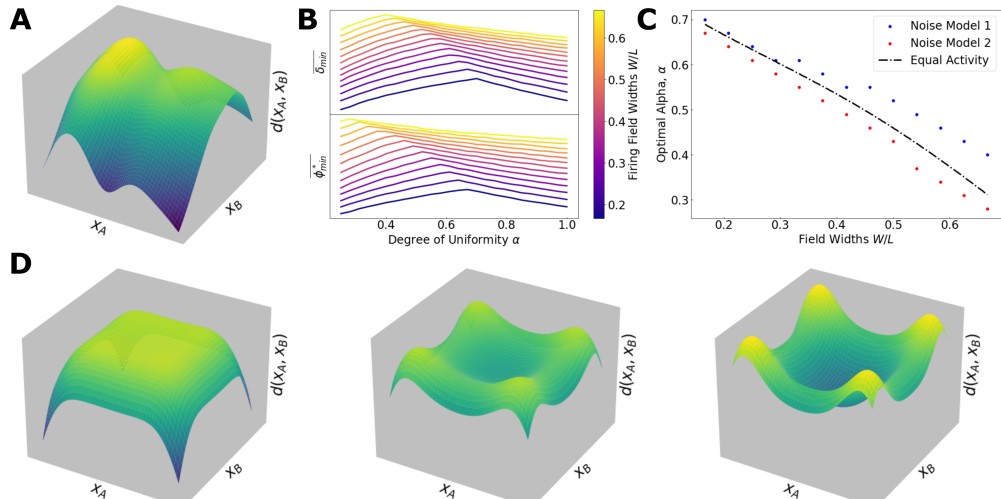

Figure 5: **(A)** An example surface swept out between distances in code space by positions $x_A$,$x_B$, $d(f_A(x_A), f_B(x_B))$. We have analagous surfaces for the rate dependent noise model. **(B)** The height, on average, of the minimum of the this surface for both noise models, and various firing field widths. **(C)** The optimal values of $\alpha$ derived from this approach, for both noise models and an approximation from equalizing firing ratios from the boundary and the bulk (**Supplemental**). **(D)** The sample to sample average of the surface removes variations due to the random choice of $f_A$. The minimum of this surface is near the boundaries at $\alpha = 1$, but jumps discontinuously the center as $\alpha$ decreases.

environment. Adjacent to a boundary, the boundary is the dominate cue and likely obscures other cues that could potentially distinguish environments. In our model, we can add an in-homogeneity to the point process for generating place field centers to increase place field density near the boundaries. For one dimensional contexts, we can parameterize this in-homogeneity via a symmetric beta distribution, $\beta(x/L; \alpha, \alpha)$ (**Supplemental**) over space. Here $\alpha$ acts as a 'uniformity' parameter. With $\alpha = 1$ we recover the homogeneous process and have uniformly distributed place cells. Values of $\alpha < 1$ will progressively bias firing field centers towards the boundaries.

To understand the effect of the bias $\alpha$ on discrimination of pairs of contexts $A$ and $B$, we can look at where the distance in rate space is smallest. These distances are given by $d(f_A(x_A), f_B(x_B))$ and $K(s(x_A, x_B), x_A, x_B)$ for each noise model, respectively. For one dimensional contexts, these can be viewed as two dimensional surfaces swept out by $x_A$ and $x_B$ (**Fig. 5A**), while for two dimensional contexts these could be viewed as four dimensional surfaces. We take the sample to sample (annealed) average first, to find where, on average, the minimum of this surface is likely to be found (**Fig. 5D**). With $\alpha = 1$ this minimum occurs most often near the boundaries of either context $A$ or $B$. As we decrease $\alpha$, the minimum of the averaged surface near the boundaries increases until it eventually jumps discontinuously to the center (**Fig. 5D**). The value of $\alpha$ for which this jump occurs is dependent on firing field width and the noise model under consideration, but is independent of $N$ at large $N$ (**Fig. 5B**). Wider firing fields tend to lead to a larger optimal bias (**Fig. 5C**). In two dimensions, we considered a distribution of firing field centers that is a product of beta distributions, $\beta(x/L; \alpha_x, \alpha_x)\beta(y/L; \alpha_y, \alpha_y)$. The analysis for the $x$ direction and $y$ direction separate, which leads to identical optimal values for the uniformity parameter $\alpha$ as in the 1-D case.

## 3  Discussion

Many researchers have proposed that the place system in the neural substrate of the cognitive map and that global place cell remapping plays a critical role in storing information about environmental context [11, 15, 16, 17, 6, 7, 8]. Further, recent work may implicate the role of place maps in general, short term memory formation [11, 37, 38, 39, 40], which might explain the need for such a large contextual capacity. In this work, we have built upon these proposals by explicitly demonstrating under realistic firing statistics that the place cell system's context storage capacity grows exponentially

with the number of neurons, and by calculating the associated exponents. This large capacity is consistent with the notion that the hippocampus is capable of pattern separating context and encoding many experiences [41, 42, 43], and here we demonstrate that a place-like coding scheme alone is sufficient in this regard. To achieve this result, we developed a geometric model of place cell activity, which allowed us to explore how this capacity changes as a function of the number of place cells in the system and of place cell firing field properties. While our strict conditions on pairwise separability leads to a coding scheme that is robust to noise, we note that less strict conditions may be more realistic, and better suit an animals behavioural needs. We primarily focused on global remapping here, but conjecture that the qualitative structure of our results remain unchanged by including the effects of partial remapping. Including these effects will give each manifold an additional width along a few dimensions due to variations that are not due to neural noise, but rather due to partial or rate remapping. Additionally, we have not considered here the complexity of decoders of the hippocampus. Although we show that context separation is achievable, the requirement of simple decoding, as well as the architecture of the underlying hippocampal network, will further constrain the contextual capacity [31, 32, 36].

We then explored implications of this model as it pertains to various objectives of the hippocampal code. In particular, we revealed a trade-off between precise encoding of local position and discrimination between different contexts. We found that tuning individual place cells for encoding local position leads to smaller place field widths, while increasing place field widths leads to improved performance for context discrimination. The size of place fields increases from the dorsal hippocampus to the ventral hippocampus [21, 24], and we suggest that this mixed population of neurons allows the hippocampus to perform both objectives efficiently. That is, our model suggests that place cells of the dorsal hippocampus are better tuned for fine grained memory, while the more widely tuned ventral cells are better tuned for pattern separation and storage of many contexts. This is consistent with experimental evidence, in which dorsal lesion typically impair spatial memory, while ventral lesions do not. Conversely, ventral lesions have been demonstrated to impair contextual memory, for example decreasing response in contextual fear experiments, but have minimal affect on spatial tasks [22, 23, 24, 25, 26].

We also found that biasing place cell centers to cluster near environmental borders improves context discrimination. Over-representation of place field activity near boundaries is well documented [27], and we predict that this bias will systematically vary across the dorsal-ventral axis of the hippocampus, with the more widely tuned ventral place cells displaying greater bias than dorsal place cells. As rodents typically explore near boundaries of an environment, the need for higher spatial resolution in these locations may also lead to a similar bias. In fact it is well established that developmental and self-organization mechanisms can produce efficient structural and functional optimizations (vision: [44, 45, 46]; audition: [47, 48, 49]; olfaction: [50, 51]; spatial cognition: [52]) and here we are suggesting that similar processes may operate in the place system.

While we have explored hippocampal codes in isolation, interactions with other spatially tuned cells, such a egocentric and allocentric border cells, likely have an effect on this bias not explored here, suggesting yet another intricate interaction between allocentric-egocentric representations in the hippocampus [53]. If place cells are implicated for general episodic memory, such an interaction may imply that boundary cells play a role in general memory. Exploring this interaction is a topic for future work, and our approach provides a foundation for exploring these avenues.

Finally, we have also focused on physical one and two dimensional contexts in this work, but our geometric formulation generalizes to higher dimensional and abstract spaces. Our derivation of the exponential scaling is independent of the dimension, and so we predict that the hippocampus should also be able to segregate context and distinguish locations in more abstract spaces efficiently. It is worth noting however that there is still an appreciable drop in performance when moving from one to two dimensional spaces, and so the system is likely incentivized to encode abstract spaces with lower dimensional structures when possible.

**Acknowledgments:** We thank Dori Derdikman, Genela Morris, and Shai Abramson for many illuminating discussions in the course of this work, which was supported in part by NIH CRCNS grant 1R01MH125544-01 and by the NSF and DoD OUSD (R&E) under Agreement PHY-2229929 (The NSF AI Institute for Artificial and Natural Intelligence). VB was supported in part by the Eastman Professorship at Balliol College, Oxford.

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

# A  Supplemental

## A.1  Simulation Distributions and Firing Fields

### A.1.1  Gamma-Poisson Distributions

The number of firing fields each place cell has is determined by a gamma-poisson distribution, as described in [29, 30]. This is a mixed poisson model, with a gamma distribution acting as the mixing distribution. That is, the rate variable of the poisson model is gamma distributed with parameters $\alpha, \beta$. We use $\alpha = 1.5$ and $\alpha = 2.25$ in one and two dimensions, respectively. We use $\beta = 4m/L$ and $\beta = 8m^2/A$ in one and two dimensions, respectively. These create distributions consistent with those seen in experiment, as below [29, 30]. The total number of neurons for generating these distributions was 200 :

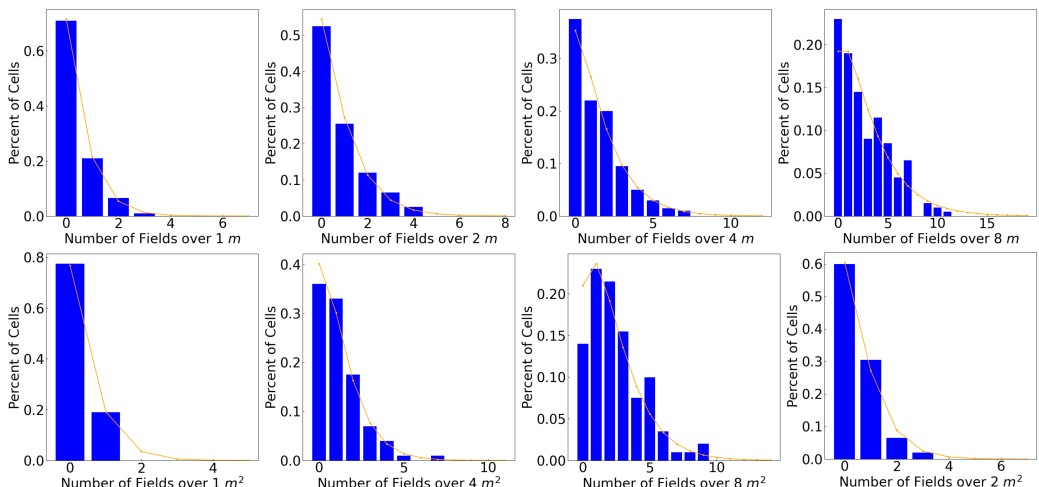

Figure 6: The number of firing fields per neuron that arise from the gamma-poisson distribution. Top is one dimensional rooms that vary in length from $1m$ to $8m$. Bottom is two dimensional rooms that vary in size from $1m^2$ to $8m^2$. As rooms get larger, neurons are recruited for representations more often

Under these statistics, most neurons are silent in small rooms, so two contexts will share few active neurons. In large rooms, few neurons have no firing fields, and so two contexts will share many active neurons.

### A.1.2  Field Definitions

For each randomly generated room, each neuron, indexed by $j$, has $a_j$ fields drawn from the gamma-poisson distribution described above. Each firing field center is placed according to a homogeneous (first two parts) or nonhomogeneous (last part) poisson process over space. Firing fields are gaussian, so that the total spatial tuning is given by a sum of gaussians:

$$f_{A,j}(x_A) = .1Hz + C_{j,A} \sum_{i}^{a_j} \exp -\frac{1}{2(w_i/2)^2}(\vec{x}_A - \vec{\mu}_{A,i})^2 \tag{20}$$

All neurons are given base firing rate of $.1Hz$, and $C_{j,A}$ is chosen so that active neurons have a max firing rate of $30Hz$ for simplicity. Max firing rates are not drawn from a distribution, to speed up pdf convergence times. Likewise, widths are not chosen stochastically, and have equal fixed width. This also speeds up pdf convergence times, and additionally lets us study the effect of changing firing field widths in isolation.

### A.1.3  High Dimensional Gaussians

Here we outline a standard method of demonstrating the exponential localization of gaussians to shells in high dimensions [33, 34]. If **z** is distributed according to a normal $N$ dimensional spherical

gaussian with variance $\sigma^2$, and $R = |\mathbf{z}|_2$ is its norm, then the distribution of $R$ is

$$p(R) = \frac{S_N R^{N-1}}{(2\pi\sigma^2)^{N/2}} \exp\{-\frac{R^2}{2\sigma^2}\} \tag{21}$$

where $S_N$ is the area of the $N$ dimensional unit sphere. Setting $\frac{\partial}{\partial R} p(R) = 0$, we find this distribution has a maximum at $\sigma\sqrt{N}$. Now for any $\sigma q << \sigma\sqrt{N}$, we have that

$$p_R(\sigma\sqrt{N} + \sigma q) = \frac{S_N(\sigma\sqrt{N} + \sigma q)^{N-1}}{(2\pi\sigma^2)^{N/2}} e^{-\frac{(\sigma\sqrt{N} + q\sigma)^2}{2\sigma^2}} \tag{22}$$

$$= \frac{S_N}{(2\pi\sigma^2)^{N/2}} e^{-\frac{N}{2} - q\sqrt{N} - \frac{q^2}{2} + (N-1)\ln(\sigma\sqrt{N} + \sigma q)} \tag{23}$$

$$= \frac{S_N}{(2\pi\sigma^2)^{N/2}} e^{-\frac{N}{2} + (N-1)\ln(\sigma\sqrt{N})} e^{-q\sqrt{N} - \frac{q^2}{2} + (N-1)\ln(1 + \frac{q}{\sqrt{N}})} \tag{24}$$

$$= p_R(\sigma\sqrt{N}) \exp\{-q\sqrt{N} - \frac{q^2}{2} + (N-1)\ln(1 + \frac{q}{\sqrt{N}})\} \tag{25}$$

$$= p_R(\sigma\sqrt{N}) \exp\{-q\sqrt{N} - \frac{q^2}{2} + (N-1)(\frac{q}{\sqrt{N}} - \frac{q^2}{2N} + \mathcal{O}(N^{-3/2}))\} \tag{26}$$

$$= p_R(\sigma\sqrt{N}) \exp\{-q^2 + \mathcal{O}(N^{-1/2})\} \tag{27}$$

And so the mass of a high dimension gaussian is exponentially localized to an annulus of radius $\sigma\sqrt{N}$. We can choose a $q$ such that the majority of the probability mass of $\mathbf{z}$ is within a a sphere of radius $\sigma(\sqrt{N} + q)$. For all the figures in the main text, we use $q = 2$, which leads to $\approx 99.8\%$ of the mass of $p(\mathbf{z})$ being within a sphere of radius $\sigma(\sqrt{N} + q)$. As $q$ is of order 1 in $N$, choosing larger values leads to a negligible effect on derived values at large $N$, which we additionally verified numerically.

For the variable noise model, $\xi \sim \mathcal{N}(0, \phi\,\mathrm{diag}\{f_A(x_A)\})$. We have then that $\xi$ at any position $x_A$ is exponentially localized to an ellipsoid with major axis of length $\sqrt{N\phi f_A^i(x_A)}$. Including the annulus width, we have that the ellipsoid width in a particular direction is $\sqrt{\phi f_A^i(x_A)}(\sqrt{N} + q)$.

### A.1.4 Ellipsoid Intersection

We want to check if the manifolds widened out by noise intersect. For the simple noise model, checking if the spheres centered at $f_A(x_A)$ and $f_B(x_B)$ for some pair $x_A, x_B$ is equivalent to checking if the distance between their centers is greater than twice their radius, or $d(f_A(x_A), f_B(x_B)) > 2\sigma(\sqrt{N} + q)$. For the variable noise model, we need an efficient way to check if two ellipsoids intersect.

From [35], two ellipsoids in a high dimensional euclidean space, defined by centers $\mu_A, \mu_B$ and covariance matrices $\Sigma_A, \Sigma_B$ intersect if and only if the convex function

$$K(s) = 1 - s\mu_a^T \Sigma_A \mu_a - (1-s)\mu_b^T \Sigma_B \mu_b + \mu_\cap^T \Sigma_s \mu_\cap \tag{28}$$

becomes negative for any choice of $s \in [0, 1]$. The proof is given in [35]. Here,

$$\Sigma_s = s\Sigma_A + (1-s)\Sigma_B \tag{29}$$

$$\mu_\cap = \Sigma_s^{-1}(s\Sigma_A \mu_a + (1-s)\Sigma_B \mu_b) \tag{30}$$

In the Poisson-like noise model, we check if the ellipsoids centered at $f_A(x_A)$, $f_B(x_B)$ in rate space intersect. These ellipsoids are given by:

$$(r - f_A(x_A))^T((\sqrt{N} + q)^2 \phi\Sigma_A(x_A))^{-1}(r - f_A(x_A)) \leq 1 \tag{31}$$

$$(r - f_B(x_B))^T((\sqrt{N} + q)^2 \phi\Sigma_B(x_B))^{-1}(r - f_B(x_B)) \leq 1 \tag{32}$$

Here, $\Sigma_A(x_A) = \mathrm{diag}[f_A(x_A)]$. I will drop the $x_A$ and $x_B$ for conciseness, with the understanding that there is an implicit dependence. To check for intersection, we plug these into the form of the convex function $K(s)$:

$$K(s, x_A, x_B) = 1 - (f_B - f_A^T)[\frac{1}{1-s}(\sqrt{N} + q)^2 \phi\Sigma_A + \frac{1}{s}(\sqrt{N} + q)^2 \phi\Sigma_B]^{-1}(f_B - f_A)$$

The ellipsoids centered at $f_A(x_A)$, $f_B(x_B)$ do not intersect if and only if there exists an $s^* \in [0,1]$ such that $K(s^*, x_A, x_B) < 0$ [35]. Rearranging, we can write this as:

$$K(s, x_A, x_B) = 1 - \frac{1}{\phi(\sqrt{N} + q)^2} \sum_i^N \frac{(f_B^i(x_B) - f_A^i(x_A))^2}{(\frac{1}{1-s} f_A^i(x_A) + \frac{1}{s} f_B^i(x_B))}$$

The choice that of $s$ that minimizes $K(s, x_A, x_B)$ changes with $x_a, x_b$, and so we can let $s^*(x_A, x_B)$ represent this value. We define $\phi^*(x_A, x_B)$ by

$$\phi^*(x_A, x_B) = \frac{1}{N} \sum_i^N \frac{(f_B^i(x_B) - f_A^i(x_A))^2}{(\frac{1}{1-s^*(x_A,x_B)} f_A^i(x_A) + \frac{1}{s^*(x_A,x_B)} f_B^i(x_B))} \tag{33}$$

This definition lets us write:

$$K(s^*(x_A, x_B), x_A, x_B) = 1 - \frac{1}{\phi(\sqrt{N} + q)^2} \phi^*(x_A, x_B) \tag{34}$$

Now we want ellipsoids centered at $f_A(x_A)$, $f_B(x_B)$ to not intersect for any $x_A$, $x_B$, which amounts to requiring that $K(s^*(x_A, x_B), x_A, x_B) < 0$ for all $x_A, x_B$, or equivalently $\max_{x_A \in A, x_B \in B} K(s^*(x_A, x_B), x_A, x_B) < 0$. To put this in a similar form as the gaussian case, we can write this as the requirement that

$$\min_{x_A, x_B} N\phi^*(x_A, x_B) > (\sqrt{N} + q)^2 \phi \tag{35}$$

## A.2  Gaussian Distributed Miminum Distances

### A.2.1  Gaussian Model

We define $\delta_{min} = \min_{x_A, x_B} d(f_A(x_A), f_B(x_B))$. $\delta_{min}$ is a function of the maps $f_A, f_B$, which are chosen at random with respect to a Gamma-Poisson distribution. Here, we find how the mean and scale of $\delta_{min}$ scale with $N$.

The distance in rate space evaluated at some pair of positions $X_A, X_B$ for large $N$ will look like the square root of the sum of the square of many random variables, as the firing fields are randomly chosen. In particular, each term in the sum itself is independent of $N$, so to get the rough shape as a function of $N$ of this distribution, we can discard some of the finer details of $f_A, f_B$.

$$d(f_A(X_A), f_B(X_B)) = \sqrt{\sum_i^N (f_A^i(X_A) - f_B^i(X_B))^2} \tag{36}$$

The construction of the firing fields of each neuron occurs independently, so $Y_i = (f_A^i(X_A) - f_B^i(X_B))^2$ represent a set of i.i.d. random variables with some mean an variance which is independent of $N$, allowing us to use the central limit theorem. We can write then

$$d(f_A(X_A), f_B(X_B)) = \sqrt{\sum_i^N (f_A^i(X_A) - f_B^i(X_B))^2} \tag{37}$$

$$= \sqrt{N\tilde{\mu}_\delta + Z_1} \tag{38}$$

$$= \sqrt{N\tilde{\mu}_\delta} \sqrt{1 + Z_2} \tag{39}$$

Where $\tilde{\mu}_\delta = \mathbb{E}[Y_i]$, and $Z_1$ is a normally distributed random variable with mean zero and variance $N\text{Var}[Y_i]$, by the central limit theorem, and $Z_2$ is normally distributed with zero mean and variance $\frac{\text{Var}[Y_i]}{N\tilde{\mu}_\delta^2}$. As $\tilde{\mu}_\delta$ is independent of $N$, at large $N$, we can expand the square root around small $Z_2$:

$$d(f_A(X_A), f_B(X_B)) \approx \sqrt{N\tilde{\mu}_\delta} + \frac{1}{2} Z_2 \tag{40}$$

We find then that the distance in rate space between $X_A$ and $X_B$ is distributed normally at large $N$, with mean scaling like $\sqrt{N}$ and variance that is independent of $N$,

$$d(f_A(X_A), f_B(X_B)) \sim \mathcal{N}(\tilde{\mu}_\delta \sqrt{N}, \tilde{\lambda}_\delta^2)$$

The constants are not determined here, though will be calculated in a simpler case later, and fit to simulations. Now $\tilde{\mu}_\delta$ and $\tilde{\lambda}_\delta$ may depend on the particular choice of positions $X_A$ and $X_B$, but this will still hold at positions where the minimum is most likely to be found, and so $P(\delta_{min})$ will also have a mean that grows with $\sqrt{N}$ and unit variance,

$$\delta_{min} \sim \mathcal{N}(\mu_\delta\sqrt{N}, \lambda_\delta^2) \tag{41}$$

for yet determined constants.

### A.2.2  Rate dependent model

We also need to calculate the behavior for $P(N\phi^*)$ at large N. We start with the definition of $\phi^*$, and want to show that it is gaussian distributed

$$N\phi^*(x_A, x_B) = \sum_i^N \frac{(f_B^i(x_B) - f_A^i(x_A))^2}{\left(\frac{1}{1-s^*(x_A,x_B)}f_A^i(x_A) + \frac{1}{s^*(x_A,x_B)}f_B^i(x_B)\right)} \tag{42}$$

Again, we consider any particular fixed $X_A$, $X_B$. Then the content of the sum can be viewed as just some random variable with respect to the distributions on $f_A$, $f_B$. The elements of the sum have non-zero mean and variance and are i.i.d. This is because in constructing neural firing fields, the placement of fields for neurons $i$ and $j$ are assumed to be independent. By the central limit theorem, we have then that

$$N\phi^*(X_A, X_B) \sim \mathcal{N}(\tilde{\mu}_\phi N, \tilde{\lambda}_\phi^2 N) \tag{43}$$

or equivalently

$$\phi^*(X_A, X_B) \sim \mathcal{N}(\tilde{\mu}_\phi, \tilde{\lambda}_\phi^2/N) \tag{44}$$

for some constants $\tilde{\mu}_\phi, \tilde{\lambda}_\phi$. The values of $\tilde{\mu}_\phi, \tilde{\lambda}_\phi$ may depend on the exact position , but will be independent of $N$ at sufficiently large $N$, and so in particular, fluctuations of $\phi^*$ will mostly be due to position. This implies that, as before, the minimum value $N\phi^*_{min}$ will have a distribution with the same scaling as $N\phi^*$ at any position, but with different constants.

$$N\phi^*_{min} \sim \mathcal{N}(\mu_\phi N, \lambda_\phi^2 N) \tag{45}$$

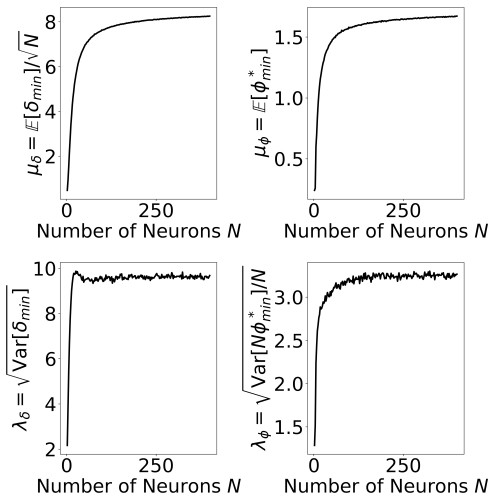

Figure 7: Numerical demonstration that the constants $\mu_\delta, \lambda_\delta, \mu_\phi, \lambda_\phi$ are independent of $N$ for large $N$ in 1-d, as predicted.

## A.3 Union Bound and Product Ansatz

In the main text, we assume that the probability that $M$ contexts are distinguishable is approximately a function of the probability that any pair is, via a saturation of the union bound $P_M = P_2^{\binom{M}{2}}$. More transparently, we assume:

$$P(M \text{ Contexts are distinguishable}) \approx \prod_{(ij)} P(i \text{ and } j \text{ are distinguishable}) = P_2^{M(M-1)/2} \quad (46)$$

Strictly speaking this is a union bound $P_M \leq P_2^{M(M-1)/2}$, as the probability that rooms 1 and 2 are distinguishable and rooms 2 and 3 are is not independent. We argue that this bound is approximately saturated at large $N$ if noise is not too strong, by demonstrating each context occupies a vanishingly small relative volume in rate space as $N$ gets large. This argument is a rough reason to think that the ansatz is true, but not an exact proof.

We consider the rate independent noise model to start. The total available volume in rate space is $\approx F_{max}^N$. How we have incorporated noise lets neuron firing rates extend past $F_{max}$, as it is a max set before noise, and so firing rates can "spill out" of this $N$ dimensional box. For one dimensional rooms, $f_A$ carves out a curve of length $l = \int dx \sqrt{\sum_i (\frac{df^i}{dx})^2}$ in rate space. For $D$ dimensional rooms, $f_A$ defines a surface, which has area

$$S = \int d^d x \sqrt{\det g} \quad (47)$$

Here $g = J_f^T J_f$, with $J_f$ being the jacobian $J_f = \partial_{x_i} f^j$. In $D$ dimensional rooms, $g$ will be a matrix, with each element scaling no faster than linearly in $N$. The determinant then can scale no faster than $N^{D/2}$, so at worst, we expect the surface area to scale in $N$ like

$$S \propto c^D N^{D/2} \quad (48)$$

where $c$ is some constant. To go from the surfaces carved out by the maps $f$ to the volumes occupied by the firing rates of the neurons, we need to consider the effect of adding noise. Now we can roughly approximate the effect of noise by putting an $N - D$ spherical cross section to each point along the surface. We avoid exact integrals as we primarily care about the scaling. The radius of each cross section is $\sigma \sqrt{N}$, and so each cross section has a volume:

$$V_{N-D}(\sigma \sqrt{N}) = \frac{\pi^{N/2}}{\Gamma(\frac{N}{2} + 1)} (\sigma \sqrt{N})^{N-D}$$

Or, using stirlings approximation to get the scaling in $N$,

$$V_{N-D}(\sigma \sqrt{N}) \sim \frac{1}{\sqrt{(N-D)\pi}} \left(\frac{2\pi e}{N-D}\right)^{N/2-D/2} (\sigma^2 N)^{N/2-D/2} \quad (49)$$

$$= \frac{1}{\sqrt{(N-D)\pi}} \left(2\pi e \sigma^2 \frac{N}{N-D}\right)^{\frac{N-D}{2}} \quad (50)$$

The approximate volume in rate space of the thickened manifolds then scales no faster in $N$ than

$$V_r \propto c^D N^{D/2} \frac{1}{\sqrt{(N-D)\pi}} \left(2\pi e \sigma^2 \frac{N}{N-D}\right)^{\frac{N-D}{2}} \quad (51)$$

Assuming the dimension of the environment $D$ is small (for physical environments, $D$ is 1, 2, or 3, but we might be interested in higher dimensional abstract 'environments') compared to $N$ and $N$ is large, then at worst the scaling is

$$V_r \propto N^{\frac{D-1}{2}} (2\pi e \sigma^2)^{\frac{N-D}{2}} \quad (52)$$

We have neglected the volume of the "caps" of these manifolds. For example, the one dimensional rooms define a tube in rate space, and have half-spherical caps on each end. The volume of the $N$ dimensional sphere of radius $\sqrt{N\sigma^2}$ is $\frac{1}{\sqrt{N\pi}} (2\pi e \sigma^2)^{N/2}$. In higher dimensions, this additional

component would scale very roughly like the length of the boundary of the surface $S$ times the volume of the $N - D + 1$ sphere.

The proportion of the volume of the thickened manifold to the total volume of rate space, $V_r / F_{max}^N$, is then vanishing in $N$ provided that the neurons are not too noisy. We get an approximate condition of the form $\sqrt{2\pi e}\sigma < F_{max}$. A more precise calculation might improve or deteriorate this bound, but interestingly this mimics typical conditions on signal to noise ratios. Each room occupies a vanishingly small fraction of the total volume in rate space as $N$ grows large. Similar arguments hold for the ellipsoidal noise model, as each ellipsoid occupies a smaller volume than its osculating sphere.

### A.4 Exponentially Many Rooms

Here, we find the scaling of the number of rooms $M$ with respect to the number of neurons $N$. We will use the product ansatz:

$$P_M = P_2^{M(M-1)/2}$$

We can invert this equation by taking the logarithm of both side, then solving the quadratic formula for $M$, to find that

$$M(N) = \sqrt{2\frac{\log(P_M)}{\log(P_2(N))} + \frac{1}{4}} + 1/2 \tag{53}$$

Here, we can let $P_M$ represent a confidence. For example, if we are okay with being 95% percent confident in storing multiple rooms, then $M(N)$ becomes an equation purely dependent on $P_2(N)$. Now $P_2(N)$ is, at large $N$, equal to an integral of a gaussian for both noise models. We start with the simpler noise model. We have that

$$P_2 = P(\delta > 2\sigma(\sqrt{N} + q)) \tag{54}$$
$$= 1 - F(2\sigma(\sqrt{N} + q)) \tag{55}$$
$$\approx 1 - \Phi(\frac{2\sigma(\sqrt{N} + q) - \mu_\delta\sqrt{N}}{\lambda_\delta}) \tag{56}$$

Here, $F$ represents the cumulative distribution, and $\Phi$ is the cumulative distribution of the standard normal distribution. The approximation comes from the central limit theorem at large $N$, with $\mu_\delta = \mathbb{E}(\delta_{min})/\sqrt{N}$ and $\lambda_\delta^2 = \text{Var}(\delta_{\min})$ are both independent of $N$. This can be written in terms of the error function for the first noise model as

$$P_{2,\delta}(N) \to \frac{1}{2}(1 - \text{erf}(\frac{2\sigma(\sqrt{N} + q) - \mu_\delta\sqrt{N}}{\sqrt{2}\lambda_\delta})) \tag{57}$$
$$= \frac{1}{2}(1 + \text{erf}((\frac{\sqrt{N}\mu_\delta - 2\sigma(\sqrt{N} + q))}{\sqrt{2}\lambda_\delta})) \tag{58}$$

We can bring the second noise model into a similar form:

$$P_{2,\phi} = P(\min_{x_A, x_B} N\phi^*(x_A, x_B) > (\sqrt{N} + q)^2\phi) \tag{59}$$
$$= 1 - F((\sqrt{N} + q)^2\phi) \tag{60}$$
$$\to 1 - \Phi(\frac{(\sqrt{N} + q)^2\phi - N\mu_\phi}{\lambda_\phi\sqrt{N}}) \tag{61}$$
$$= \frac{1}{2}(1 - \text{erf}(\frac{(\sqrt{N} + q)^2\phi - N\mu_\phi}{\sqrt{2}\lambda_\phi\sqrt{N}})) \tag{62}$$
$$= \frac{1}{2}(1 + \text{erf}(\frac{N\mu_\phi - (\sqrt{N} + q)^2\phi}{\sqrt{2}\lambda_\phi\sqrt{N}})) \tag{63}$$

Here, $F$ is the cumulative distribution for $\phi_{min}^*N$, $\mu_\phi = \mathbb{E}(\phi_{min}^*)/N$ and $\lambda_\phi^2 = \text{Var}(\phi_{\min}^*)/N$ are both independent of $N$.

In both noise models, we can write $P_2 = \frac{1}{2}(1 + \text{erf}(u))$, with $u$ given in equation 58 and 63 for the two noise models, respectively. $P_2$ approaches either 1 or 0 for large $N$, depending on the strength of the noise with respect to the mean distance $\mu$. We have then:

$$M(N) = \sqrt{2\frac{\log(P_M)}{\log(\frac{1}{2}(1 + \text{erf}(u)))} + \frac{1}{4}} + 1/2 \tag{64}$$

In the noisy regime, this approaches 1 at large $N$. In the weak to moderate noise regime, we have that $u$ approaches $\infty$ as $N$ approaches $\infty$ in both models, and the error function approaches 1. We are interested in its asymptotic behaviour. We can use the following expansion at large $u$:

$$\text{erf}u = 1 - \frac{e^{-u^2}}{\sqrt{\pi}}(u^{-1} - \frac{1}{2}u^{-3} + ...) \tag{65}$$

Expanding to leading order in $u$ then gives:

$$M(N) \approx \sqrt{2\frac{\log(P_M)}{\log(1 - \frac{1}{2}\frac{\exp(-u^2)}{\sqrt{\pi}}(u^{-1} + \mathcal{O}(u^{-3})))}} \tag{66}$$

Now we can expand the logarithm by noticing that for small values,

$$\sqrt{-\frac{A}{\ln(1-x)} + B} \approx \sqrt{\frac{A}{x}} + \mathcal{O}(x^{\frac{1}{2}}) \tag{67}$$

So to leading order, we have:

$$M(N) \approx \sqrt{-2\frac{\log(1/P_M)}{\log(1 - \frac{1}{2}\frac{\exp(-u^2)}{\sqrt{\pi}}u^{-1})}} \tag{68}$$

$$\approx \sqrt{2\log 1/P_m}[\frac{1}{2}\frac{\exp(-u^2)}{\sqrt{\pi}}u^{-1}]^{-1/2} \tag{69}$$

$$= 2\sqrt{\sqrt{\pi}\log(1/P_m)}\sqrt{u}e^{\frac{1}{2}u^2} \tag{70}$$

At large $N$, the constant $q$ is negligible, and so we can drop it, giving $u$ for the rate independent and the rate dependent model as $u = \frac{\mu_\delta - 2\sigma}{\sqrt{2}\lambda_\delta}\sqrt{N}$, and $u = \frac{\mu_\phi - \phi}{\sqrt{2}\lambda_\phi}\sqrt{N}$, respectively. Plugging in for both noise models gives

$$\text{Model 1: } M(N) \approx \sqrt{\ln(1/P_M)\frac{\mu_\delta - 2\sigma}{\lambda_\delta}}(8\pi N)^{1/4}\exp\left[\frac{1}{4}(\frac{\mu_\delta - 2\sigma}{\lambda_\delta})^2 N\right] \tag{71}$$

$$\text{Model 2: } M(N) \approx \sqrt{\ln(1/P_M)\frac{\mu_\phi - \phi}{\lambda_\phi}}(8\pi N)^{1/4}\exp\left[\frac{1}{4}(\frac{\mu_\phi - \phi}{\lambda_\phi})^2 N\right] \tag{72}$$

In both noise models, the number of storable contexts as a function of $N$ scales like

$$M(N) \sim N^{1/4}e^{\gamma N} \tag{73}$$

with the constant $\gamma$ in each noise model being given by

$$\gamma_\delta = (\frac{\mathbb{E}[\delta_{min}]/\sqrt{N} - 2\sigma}{2\sqrt{\text{Var}(\delta_{min})}})^2 \qquad \gamma_\phi = (\frac{\mathbb{E}[\phi^*_{min}] - \phi}{2\sqrt{N\text{Var}([\phi^*_{min}])}})^2 \tag{74}$$

These values are shown for various noise levels, firing field widths, and room sizes in the main text figure 3 and the supplemental figure 8.

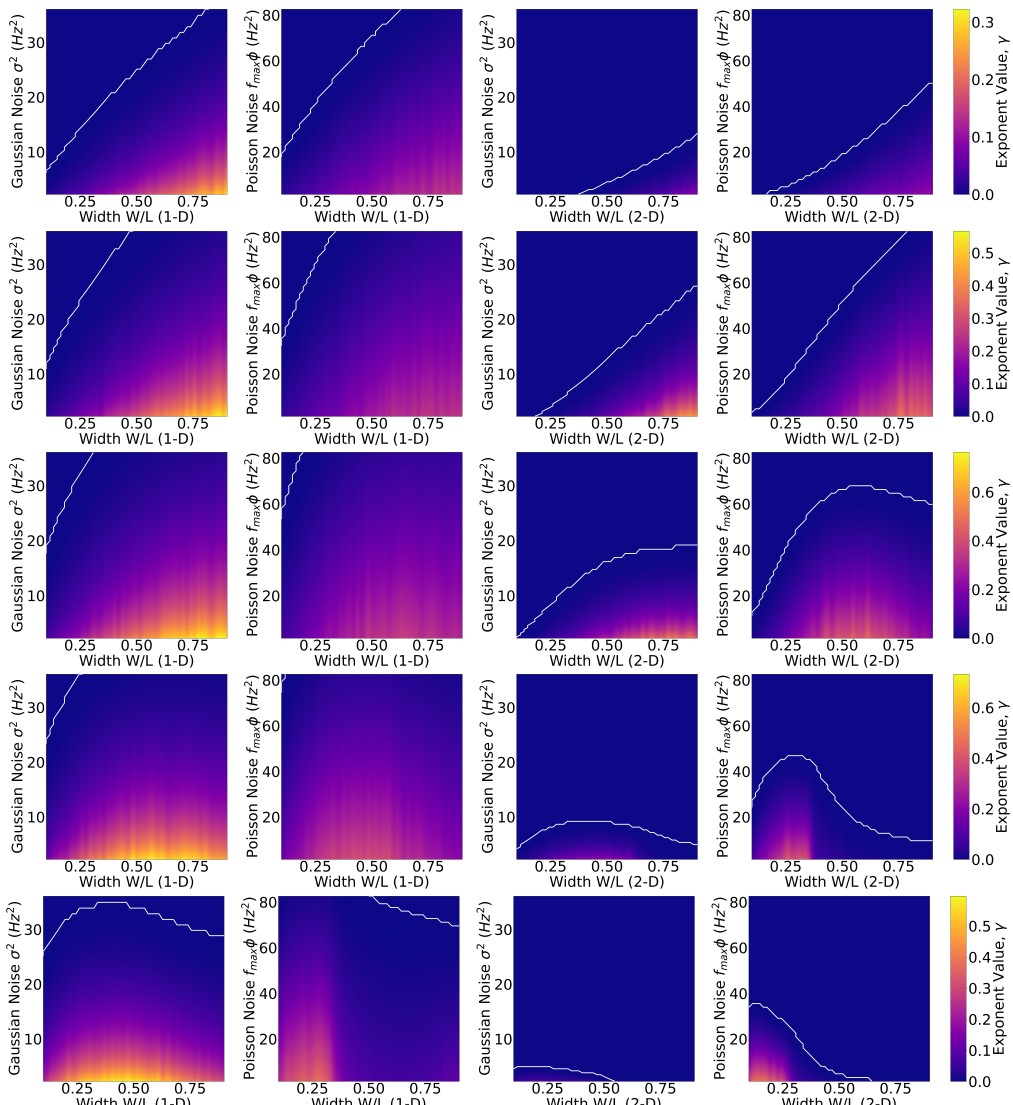

Figure 8: The value of the exponential $\gamma$ at large $N$ as a function of firing field width and noise. From left to right are 1d rooms, Gaussian noise; 1d rooms, Poisson-like noise; 2d rooms, Gaussian noise; and 2d rooms, Poisson-like noise. Rooms increase in size from top to bottom ($1m$, $2m$, $4m$, $8m$, and $16m$ rooms, and $1m^2$, $2m^2$, $4m^2$, $8m^2$, and $16m^2$ rooms). Under our Poisson Gamma statistics for the number of active fields, we expect in smaller rooms, where the number of active fields is small, to preference large fields for the objective of context segragation. Conversely, in large rooms, extremely large fields become harmful for context segragation. White demarcates the region where decoding multiple contexts is possible. The Poisson-like model is generally much more robust to noise.

### A.4.1 Infinite Width Derivations

For neurons with infinite width, the number of storable contexts is purely a function of the number of neurons which have active fields. In this limit, if there are $N$ neurons, each context gets a binary identifier. Consider two contexts $A$ and $B$. If each neuron has a probability $\theta$ of being active, the distribution of the hamming distance between these two contexts is given by

$$P(H = k) = \binom{N}{k} [\theta^2 + (1-\theta)^2]^{N-k} [2\theta(1-\theta)]^k \tag{75}$$

Now we are interested in $P(\min d(f_A(x_A), f_B(x_B)) > 2\sigma\sqrt{N})$. In the infinite width limit, the spatial depedence is lost, and $d \to f_{max}\sqrt{H}$. We can write then that the probability that two rooms are separable is

$$P(H > \frac{4\sigma^2}{f_{max}^2} N) = \sum_{k=\lceil \frac{4\sigma^2}{f_{max}^2} N \rceil}^{N} P(H = k) \tag{76}$$

The distribution $P(H)$ is well approximated by a gaussian at large $N$, with mean $\mu = 2Nq(1-q)$ and variance $\sigma^2 = 2N(1-\theta)\theta[(1-\theta)^2 + \theta]$. From this, we get an integral similar to the one arrived at in the main text. At large $N$, the probability that two rooms are separable approaches one if $2\theta(1-\theta) > \frac{4\sigma^2}{f_{max}^2}$. Rooms are optimally separable in the infinite width limit when $\theta = 1/2$.

## A.5 The Cramer-Rao Bound

To quantify a population level code for position, we consider estimators $\hat{x}$ of position constructed from population level firing, and compare to true position. In one dimension, the quality of any estimator $\hat{x}$ at a particular position is bounded below by the inverse Fisher information:

$$\langle (\hat{x} - x)^2 \rangle_{r|x} = \text{Var}(\hat{x}) \geq \mathcal{I}^{-1}(x) \tag{77}$$

In one dimension, the Fisher metric is given by:

$$\mathcal{I} = \mathbb{E}_r[(\frac{\partial}{\partial x} l)^2] \tag{78}$$

Where $l$ is the log likelihood of firing rates given position.

In higher dimensions, this Cramer-Rao bound is a statement about covariance:

$$\langle (\hat{x} - x)(\hat{x} - x)^T \rangle \geq \mathcal{I}^{-1}(x) \tag{79}$$

Here, $\geq$ indicates that the difference of the LH and RH side is positive semi-definite. To get the mean-squared error, we need to take a trace over the inverse Fisher information metric:

$$\langle (\hat{x} - x)^2 \rangle = \text{Tr} \langle (\hat{x} - x)(\hat{x} - x)^T \rangle \tag{80}$$

$$= \text{Tr}\,\text{Cov}(\hat{x}) \geq \text{Tr}(\mathcal{I}^{-1}(x)) \tag{81}$$

In higher dimensions, $\text{Cov}(\hat{x}) \geq \mathcal{I}^{-1}(x)$ is a statement about the difference being positive semi-definite, which leads to our trace condition above. The Fisher Metric is given by:

$$\mathcal{I} = n\mathbb{E}_r[(\nabla_x l)(\nabla_x l)^T] = -n\mathbb{E}_r[\nabla_x \nabla_x^T l] \tag{82}$$

The averages in the Cramer-Rao bound are taken with respect to firing rates given positions. As we are interested in having good spatial resolution across positions and contexts, we additionally take averages over both. Below, we plug in the log-likelihoods for both noise models.

### A.5.1 Gaussian Noise

In 1 dimension, we have that

$$l = \sum_i \ln P(r_i|x) = -\sum_i \frac{1}{2\sigma^2}(r_i - f_A^i(x))^2 \tag{83}$$

The Fisher Information then is given by

$$\mathcal{I} = n\mathbb{E}_r[(\frac{\partial}{\partial x}l)^2] \tag{84}$$

$$= \frac{1}{\sigma^4}\mathbb{E}_r[\sum_i f_A^{i\,\prime}(x)(r_i - f_A^i(x)) \sum_j f_A^{j\,\prime}(x)(r_j - f_A^j(x))] \tag{85}$$

We have that

$$\mathbb{E}_r[r_i r_j] = f_A^i(x)f_A^j(x) + \delta_{ij}\sigma^2 \quad ; \quad \mathbb{E}_r[r_i] = f_A^i(x) \tag{86}$$

So the fisher information is

$$\mathcal{I} = n\mathbb{E}_r[(\frac{\partial}{\partial x}l)^2] \tag{87}$$

$$= \frac{1}{\sigma^4}\mathbb{E}_r[\sum_{i,j} f_A^{i\,\prime}(x)f_A^{j\,\prime}(x)(r_i r_j - r_j f_A^i(x) - r_i f_A^j(x) + f_A^i(x)f_A^j(x))] \tag{88}$$

$$= \frac{1}{\sigma^4}[\sum_{i,j} f_A^{i\,\prime}(x)f_A^{j\,\prime}(x)(f_A^j(x)f_A^i(x) + \sigma^2\delta_{ij} - 2f_A^j(x)f_A^i(x) + f_A^i(x)f_A^j(x))] \tag{89}$$

$$= \frac{1}{\sigma^2}[\sum_i f_A^{i\,\prime}(x)^2] \tag{90}$$

The CR bound will be small when this is large. This implies that, the wider the firing fields, the smaller the derivatives of $f_A^i$ will be at most locations, which leads to a smaller Fisher Information, and so a worse bound. This leads to typically narrow fields. However, if we are in a region where all firing fields are zero, the FI vanishes, and the bound explodes. This implies that the widths shouldn't be so small that there are regions with no firing fields. We also have that as the noise increase, the information decreases, and the bound gets worse, as we expect.

In 2d, we replace derivatives with divergences, and take a trace at the end. We have:

$$\nabla_x l = -\nabla_x \sum_i \frac{1}{2\sigma^2}(r_i - f_A^i(x))^2 \tag{91}$$

$$= -\sum_i \frac{1}{\sigma^2}(r_i - f_A^i(x))\nabla f_A^i(x) \tag{92}$$

Going through the same steps as before, we find that we just replace the ordinary product from before with an outer product:

$$\mathcal{I} = \frac{1}{\sigma^2}[\sum_i (\nabla f_A^i(x))(\nabla f_A^i(x))^T] \tag{93}$$

### A.5.2 Rate Dependent Noise

In the second noise model,

$$P(r_i|x) = \frac{1}{\sqrt{2\pi\phi f_A^i(x)}} \exp - \frac{1}{2\phi f_A^i(x)}(r_i - f_A^i(x))^2 \tag{94}$$

Our log likelihood is

$$l = \sum_i [-\frac{1}{2\phi f_A^i(x)}(r_i - f_A^i(x))^2 - \ln\sqrt{2\pi\phi f_A^i(x)}] \tag{95}$$

$$= \sum_i [-\frac{1}{2\phi f_A^i(x)}(r_i - f_A^i(x))^2 - \frac{1}{2}\ln f_A^i(x)] + ... \tag{96}$$

This time, we will start with divergences, and demote them for 1d rooms. We have:

$$\nabla_x l = \sum_i [-\frac{1}{\phi f_A^i}(r_i - f_A^i)\nabla f_A^i + \frac{1}{2\phi f_A^{i2}}(r_i - f_A^i)^2 \nabla f_A^i - \frac{1}{2f_A^i}\nabla f_A^i] \tag{97}$$

$$= \sum_i [-\frac{(f_A^i(x)^2 + \phi f_A^i(x) - r_i^2)}{2\phi f_A^i(x)^2}\nabla f_A^i] \tag{98}$$

The Fisher Information then is

$$\sum_{ij} \mathbb{E}_r[(\frac{(f_A^i(x)^2 + \phi f_A^i(x) - r_i^2)}{2\phi f_A^i(x)^2})(\frac{(f_A^j(x)^2 + \phi f_A^j(x) - r_j^2)}{2\phi f_A^j(x)^2})]\nabla f_A^i \otimes \nabla f_A^j \tag{99}$$

We have that

$$\mathbb{E}_r[r_i^2] = f_A^i(x)^2 + \phi f_A^i(x) \tag{100}$$

$$\mathbb{E}_r[r_i^2 r_j^2]|_{i\neq j} = (f_A^i(x)^2 + \phi f_A^i(x))(f_A^j(x)^2 + \phi f_A^j(x)) \tag{101}$$

$$\mathbb{E}_r[r_i^4] = f_A^i(x)^4 + 3\phi^2 f_A^i(x)^2 + 6f_A^i(x)^2 \phi f_A^i(x)^2 \tag{102}$$

$$= (f_A^i(x)^2 + \phi f_A^i(x))(f_A^i(x)^2 + \phi f_A^i(x)) + 4\phi f_A^i(x)^3 + 2\phi^2 f_A^i(x)^2 \tag{103}$$

$$\mathbb{E}_r[r_i^2 r_j^2] = (f_A^i(x)^2 + \phi f_A^i(x))(f_A^j(x)^2 + \phi f_A^j(x)) + \delta_{ij}[4\phi f_A^i(x)^3 + 2\phi^2 f_A^i(x)^2] \tag{104}$$

$$\tag{105}$$

Most terms in the expectation cancel, except for where $i = j$:

$$\mathcal{I} = \sum_i (\frac{4\phi f_A^i(x)^3 + 2\phi^2 f_A^i(x)^2}{(2\phi f_A^i(x)^2)^2}\nabla f_A^i \otimes \nabla f_A^i \tag{106}$$

$$= \sum_i (\frac{1}{\phi f_A^i(x)} + \frac{1}{2f_A^i(x)^2})\nabla f_A^i \otimes \nabla f_A^i \tag{107}$$

In 1-d, we replace the outer product and the divergences with ordinary scalar product and derivatives, respectively.

## A.6 Biasing Towards Boundaries

We investigated the effect of biasing place cell centers towards the boundary of environments. In 1-d, we bias the place cell centers toward boundaries by drawing centers from a symmetric beta distribution, so that the bias can be characterized by the 1 parameter family of distributions:

$$P(\mu) = \beta(\mu/L; \alpha, \alpha) = \frac{\Gamma(2\alpha)}{\Gamma(\alpha)^2}(\mu/L)^{\alpha-1}(1 - \mu/L)^{\alpha-1} \tag{108}$$

We expect an improvement in decoding multiple contexts when the total squared activity near the bulk and near the center is, on average, equal within a context. In one dimension, for an environment of length $L$, we want then approximate equality:

$$\langle\sum_i f_i(0)^2\rangle_A = \langle\sum_i f_i(L/2)^2\rangle_A \tag{109}$$

The average is with respect to number of firing fields centers and their location. If we assume for simplicity that each neuron has exactly one firing field, then we have equality when

$$\langle f_{max}^2 e^{-\frac{4}{w^2}\mu^2}\rangle_\mu = \langle f_{max}^2 e^{-\frac{4}{w^2}(\mu-L/2)^2}\rangle \tag{110}$$

Taking the average with respect to the beta distribution, the optimal bias can be found then by finding where the following integral is zero:

$$\frac{\Gamma(2\alpha)}{\Gamma(\alpha)^2}\int_0^L d\mu(\mu/L)^{\alpha-1}(1 - \mu/L)^{\alpha-1}(e^{-\frac{4}{w^2}\mu^2} - e^{-\frac{4}{w^2}(\mu-L/2)^2}) \tag{111}$$

Or,

$$\int_0^1 dx x^{\alpha-1}(1 - x)^{\alpha-1}(e^{-\frac{4L^2}{w^2}x} - e^{-\frac{4L^2}{w^2}(x-1/2)^2}) = 0 \tag{112}$$

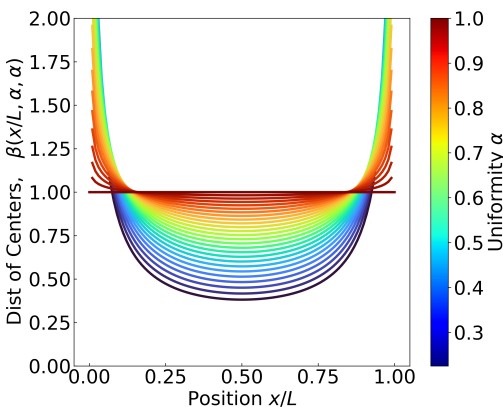

Figure 9: Beta distribution, from which we draw place cell centers. $\alpha$ is a degree of uniformity, with $\alpha = 1$ reproducing the uniform distribution.

To make this calculation numerically stable, we do the integral over an interior region (small $\epsilon$)

$$\int_{\epsilon}^{1-\epsilon} dx \, x^{\alpha-1}(1-x)^{\alpha-1}\left(e^{-\frac{4L^2}{w^2}x} - e^{-\frac{4L^2}{w^2}(x-1/2)^2}\right) = 0 \tag{113}$$

We can invert this numerically to get an approximation of the optimal $\alpha$ based on equalization of firing densities in the bulk and at the boundary. We find decent agreement between this rough calculation and the true optimal bias found in the main text for both noise models, slightly overestimating and underestimating the rate independent and the rate dependent model, respectively (Fig 5 C of the Main Text).

## A.7 Additional Computational Details

To reconstruct probability distributions for distances, we generate a large number of pairs of rooms with a fixed number of neurons and widths, calculate distances in rate space, and then find the one dimensional distributions over distances by taking a kernel density estimate over these generated distances. Although the space of possible curves in our rate space is large, since we only need to reconstruct the 1 dimensional distributions in $\delta_{min}$ and $\phi_{min}$, which are asymptotically gaussian, sampling this space suffices. We create between 1000 and 10000 samples in each numerical experiment. To speed up calculations, we perform the sampling of surfaces in parallel using PyTorch [54]. We then use scikit learns's built in Kernel density estimator with a gaussian kernel to reproduce the distributions in $\delta_{min}$ and $\phi_{min}$ [55]. As the reconstructed distribution can depend heavily on the choice of bandwidth used for the kernel, we calculate the kernel density multiple times for various choices of the bandwidth, and pick the best bandwidth via 5-fold cross validation. All code was run on a machine with an Intel Xeon-2145 processor and with a Titan Xp GPU for parallelized workflows.

