# OpenReview forum: "Trading Place for Space: Increasing Location Resolution Reduces Contextual Capacity in Hippocampal Codes"
_NeurIPS.cc/2024/Conference — NeurIPS 2024 oral_

### Official Review · Reviewer_DtE5 · 2024-07-05

**Soundness:** 4
**Presentation:** 4
**Contribution:** 4
**Rating:** 8
**Confidence:** 4

**Summary:**

This paper develops an analytical framework for understanding the coding of space and context in the hippocampus. The novel contributions include a characterization of how tuning width contributes to these functions and the trade-off between them. In particular, smaller tuning widths improve spatial localization but impair context discrimination. The authors suggest that this explains the functional role played by the gradient of tuning widths along the dorsal-ventral axis of the hippocampus. The model also explains why place cells might cluster near boundaries.

**Strengths:**

- The paper addresses an important set of issues within systems neuroscience in a novel way. The hippocampus has long been implicated in both spatial and contextual coding, but the relationship between these has not been elucidated so systematically. I believe this could have a potentially large impact, at least within the community of theorists.

- The paper is clearly written.

- The analysis is rigorous.

- The paper makes some interesting experimental predictions (though see below for connection to existing experimental work).

**Weaknesses:**

Overall my critical comments are relatively minor (see below). I do have one major comment pertaining to the model's empirical predictions. The paper makes an interesting and testable prediction that the more widely tuned cells in the dorsal hippocampus are specialized for context discrimination, whereas the more narrowly tuned cells in the ventral hippocampus are specialized for fine-grained spatial discrimination. First, I want to point out that this is backwards: ventral cells have wider tuning than dorsal cells. The classic study of this is Kjelstrup et al. (2008, Science), not cited here (see also Komorowski et al., 2013, Journal of Neuroscience). Oddly, the authors cite two papers to support their claim about the dorsal-ventral axis (Lee et al., 2020; Tanni et al., 2022), neither of which actually support this claim. The Lee et al. paper only measures activity in dorsal cells, and it's not clear which subregion was measured in the Tanni paper.

The authors propose selective inhibition experiments to test these predictions. In fact, such experiments have been done, and unfortunately they don't consistently support the predictions (none of the studies mentioned below are cited in the paper). The model would make more sense in light of at least some of these studies if the dorsal/ventral division of labor was reversed from what the authors proposed, consistent with the electrophysiology data. The review by Fanselow & Dong (2010, Neuron) provides a more systematic discussion of studies dissociating dorsal and ventral subregions.
- Richmond et al. (1999, Behavioral Neuroscience) showed that ventral lesions actually *improve* water-maze performance (a classic test of fine-grained spatial memory), whereas dorsal lesions apparently have no effect. Ventral lesions also impaired contextual fear conditioning, but dorsal lesions apparently had no effect except in the test phase where they *increased* conditioned responding to context. See also Bannerman et al. (2003, Behavioural Brain Research) for related results.
- Hock & Bunsey (1998, Journal of Neuroscience) showed that dorsal, but not ventral, lesions impaired performance on a spatial delayed alternation task, which requires memory of actions in particular spatial locations.
- On the other hand, Moser et al. (1995, PNAS) showed that dorsal lesions selectively impair spatial memory, whereas ventral lesions do not.

The authors predict that place cells should cluster near boundaries to support context segregation. No studies are cited to support this prediction, but this is something that has been studied. Consistent with the model prediction, place fields tend to occur near boundaries (e.g., Wiener et al., 1989, Journal of Neuroscience; Hetherington & Shapiro, 1997, Behavioral Neuroscience).

Minor:

- "hippocampus" is inconsistently upper-case/lower-case. I think it should be lower-case.

- p. 3: Eq. 1 should have brackets around the exponentiated term.

- p. 5: "dominate" -> "dominant" [also p. 8]

- p. 6: "an increase the distance" -> "an increase in the distance"

- p. 6: "severally" -> "severely"

- p. 9: "environmnet" -> "environment"

**Questions:**

- Can the authors do a better job relating their work to existing literature (see Weaknesses)? I understand that due to space limitations it is unlikely that they will be able to comprehensively address this literature, but I want to make sure that the work is at least largely in alignment with what is known.

- The predictions depend on noise level. Is this something that can be tested experimentally using firing rate variability?

**Limitations:**

The authors briefly discuss some modeling limitations. There are no negative societal effects of this work.

---

> ### Author Rebuttal · Authors · 2024-08-06
>
> We are grateful for the positive response, as well as the valuable suggestions and pointers to existing literature.
>
> In the initial submission, there was a  typo that propagated throughout our text, exchanging dorsal and ventral, and hence also the predicted scaling along the dorso-ventral axis. We thank the reviewer for pointing this out! In fact the ventral cells have wider tuning then the dorsal cells, and we will fix this typo in the final paper, and include citations to the Kjelstrup et al. and the Komorowski et al papers.  Note that our predictions are based more on the relative firing field sizes of place cells, and not on where these cells are located, so the key results of our paper remain unchanged. In particular, we predict that cells with wider tunings are better tuned for contextual separation but are worse at fine tuned spatial tasks, while those with narrower tunings are better tuned for tasks associated with fine-grained memory, but worse at contextual separation (though both are still able to contribute to both tasks). Correcting our dorso-ventral typo, our suggestions about experiments involving the dorsal hippocampus should be replaced with experiments involving the ventral hippocampus, and vice-versa. In particular, as the ventral cells are those with wider tuning, the corrected prediction is that selective inhibition of ventral cells should lead to worse contextual performance. Conversely, the fact that the more dorsal cells have narrower tuning suggests that selective inhibition of the dorsal hippocampus should lead to worse performance in fine grained memory tasks under our model. After this correction, our model is in line with the experimental papers you mention. In particular, our theory is supported by the result that lesions of the dorsal hippocampus impair spatial tasks (Moser et. al. 1995, Hock and Bunsey 1998) while lesions of the ventral hippocampus do not meaningfully affect performance on spatial tasks, and we will cite these experiments in the final paper. Likewise, the fact that conditioned contextual fear responses, like those described in Richmond et. al. (1999) and Bannerman et. al. (2003), are impaired after ventral lesions also supports our hypothesis that the more widely tuned neurons are better for context separation.
>
> As for the predictions involving clustering near boundaries, we will likewise cite the relevant experimental work in the final paper. In particular, the increased incidence of place cells near boundaries in Wiener et al. (1989) is in line with the predictions made by our model. As for noise variability experiments, one could inject white noise into hippocampal neurons through electrical input, or pharmacologically increase firing variability, which under our model should reduce both spatial and contextual specificity. In particular, there is a noise threshold above which the ability to separate context disappears in our geometric model. However our condition for contextual separability is likely stricter than one implemented by the hippocampus of a realistic animal, so contextual separation should persist to some extent past this noise threshold – i.e., we expect the threshold to be more of a smooth crossover than a sharp transition.
>
> Finally we will address the minor edits suggested by the reviewer.

---

> > ### Comment · Reviewer_DtE5 · 2024-08-12
> > **response to rebuttal**
> >
> > Thanks for addressing my comments. I'm glad to see that the model is more consistent with existing data than I thought. I will maintain my already high score.

---

> > > ### Author Response · Authors · 2024-08-13
> > > **thank you**
> > >
> > > Thank you for your remarks about our paper and the response.  We are grateful for your positive evaluation.

---

### Official Review · Reviewer_V9KM · 2024-07-12

**Soundness:** 3
**Presentation:** 3
**Contribution:** 2
**Rating:** 7
**Confidence:** 2

**Summary:**

This paper offers a computational investigation on the problem of encoding environmental information using population codes based on place cells, which are known to play a key role in hippocampal encoding of context, experience / goals and spatial locations. The authors propose to analyze the geometry of hippocampal codes, with the aim of precisely quantifying the capacity and properties of context encoding by place cells with different firing properties. Their analysis reveals that the number of storable contexts (i.e., strictly separable manifolds) grows exponentially with the number of place cells, showing that the hippocampus might in fact have an exponential storing capacity under realistic firing statistics of place cells.

**Strengths:**

I think that this work is a nice example of a theoretical contribution in the field of computational neuroscience. I am not familiar enough with the related literature to evaluate the originality of the approach, but from my understanding the analyses are well-conducted and well-motivated. The paper is written in a clear way.

**Weaknesses:**

The main issue with this submission, according to my non-expert opinion, is that it might have a limited relevance (and impact) on the NeurIPS community. Indeed, although I am aware that NeurIPS welcomes contributions more focused on neuroscientific aspects of neuronal computation, there is not a single NeurIPS paper cited in the literature, suggesting that this type of work might be more appropriate for other venues.

**Questions:**

-	Please always explicitly describe the variables used in equations (also in figure captions).
-	“To determine whether two contexts manifolds are separable, we use a strict criterion: the two manifolds are separable if and only if they do not have any intersections.”. This requirement seems quite strong, because it assumes that we need to decode context from any position in the manifold… Could it be replaced by a smoother criterion and/or by some form of probabilistic (linear) discriminability?
-	“We postulate that selective inhibition along the hippocampus will lead to different types of memory impairment for spatial tasks”. How would be possible to test this hypothesis experimentally? The authors mention “performing confusion experiments” but it would be interesting to better discuss how such experiments would look like.
-	Please always explicitly state where the information can be found in the Supplemental material.
-	Line 157: determine

**Limitations:**

The authors have discussed possible limitations of their work, though not in a totally explicit way.

---

> ### Author Rebuttal · Authors · 2024-08-06
>
> We appreciate the time taken by the reviewer to review our submission, and the suggestions provided.
>
> We believe that our submission is relevant to the NeurIPS community, and in particular, to the neural coding section of the Neuroscience topic, which is listed in the call for submissions. As contextual discriminability and spatial memory are both implicated by hippocampal function, approaching both through the geometry of the underlying hippocampal codes will be of interest to this community. Although the work presented here does not directly touch on the applications or theory of artificial networks, we believe that a better understanding of biological neural networks will lead to a better understanding of artificial networks, and give insights into how to design them for certain tasks. For instance, research at DeepMind (Banino et. al, 2018, Nature), by Cueva and Wei (ICLR 2018)  and by Sorscher et al (NeurIPS 2019) has found that grid-like representations, similar to those found in the Medial Entorhinal Cortex, emerge naturally for networks trained on spatial navigation, leading to deep learning agents with mammal-like navigational abilities. Likewise, place cells in the hippocampus are implicated in general short term memory (Benna and Fusi, 2020, PNAS). So a better theoretical understanding of hippocampal function will be of interest to the wider NeurIPS community when it comes to designing networks capable of flexible memory storage and retrieval.
>
> The reviewer asked if we could have used alternate criteria for separation of neural manifolds. Indeed, there are some options. For example, the work of SueYeon Chung involves a linear separation criterion for perceptual manifolds in deep neural networks (Chung et. al., 2018 APS, Chung et. al, 2020, Nature Communications). Pursuing a simpler separability criteria between activity manifolds, like the linear separation of point-cloud manifolds used in the above work, but in the context of hippocampal coding, is a possible direction we would like to explore in future research.
>
> As for selective inhibition experiments, these are possible via induced lesions to various regions of the hippocampus in rodents.  Lesions in the dorsal region of the hippocampus should lead to impairment on tasks in which high spatial resolution is crucial, such as during maze navigation, while lesions to the ventral hippocampus should lead to greater impairment in contextual tasks. (A typo in our text that inverted dorsal and ventral, but we will correct this in the final version.  Also see comments and responses to reviewer Dte5.) Some of these experiments have already been performed (again see comments to Dte5), and are in agreement with our (typo-corrected) predictions. We will also expand on the possible confusion experiments that could be run to test our hypothesis, and include the relevant citations. With regard to explicit descriptions for variable names and references to the supplemental material, we will make these both more clear in the final submission as well.

---

> > ### Comment · Reviewer_V9KM · 2024-08-08
> >
> > I thank the Authors for having considered my comments. After reading their answers and the opinion of the other Reviewers, I am  persuaded that this work could be of interest for the deep learning / neural computation communities (though maybe only partially to the NeurIPS community at large). I therefore raised my score from 6 to 7.

---

> > > ### Author Response · Authors · 2024-08-13
> > > **thank you**
> > >
> > > Thank you so much for taking the time to read our paper and the responses.  We are grateful for the improved score.

---

### Official Review · Reviewer_FBYZ · 2024-07-12

**Soundness:** 3
**Presentation:** 3
**Contribution:** 3
**Rating:** 6
**Confidence:** 4

**Summary:**

In this paper, the authors take a geometric approach of analysing context-encoding capacities admitted by place cells population firing. Specifically, through examining the manifolds underlying neural activities within different environments, the authors propose to quantify the separability of context encoding based on the overlap between the manifolds. Somewhat surprisingly, the authors noted a tradeoff between spatial specificity and contextual separation. Under the context separation constraints, the resulting place cells are tuned to be densely distributed around boundaries, which is a useful testable experimental prediction.

**Strengths:**

- The paper is well written, with clear pointers to mathematical details where appropriate.
- The tradeoff between spatial specificity and context separability is novel and sounding.
- The proposed geometric analysis is a novel framework for studying the nature of contextual representations in place cells.

**Weaknesses:**

- The paper only addresses global remapping, and did not study the implications of proposed model in terms of partial or rate remapping.

**Questions:**

See weaknesses.

**Limitations:**

See weaknesses.

---

> ### Author Rebuttal · Authors · 2024-08-06
>
> Thank you for the comments and for taking the time to review our paper. Indeed, we chose to focus our paper on global remapping, with only brief discussion of rate and partial remapping. This is because global remapping has a more dramatic and constraining effect than rate/partial remapping in our analysis: in terms of our manifold picture, partial and rate remapping alter or deform the existing manifolds, while global remapping involves “jumping” from one manifold to another. Global remapping often occurs in situations where context shifts dramatically, which for example can occur when animals move, or are moved, from one environment to another, even if the environments are superficially similar (Leutgeb et. al., 2005 Science, Alme et. al. 2014, PNAS). Partial and rate remapping often occur when an environment is modified, such as via slight changes to wall geometry, introduction of olfactory cues, or movement of cue cards within the same environment (Leutgeb et. al. 2005 Science, Bostock et. al. 1991 Hippocampus, Anderson and Jeffrey, 2003 JNeurosci). In our view, these experiments demonstrate that partial and rate remapping involve a “deformation” of existing memory tasks, while global remapping occurs and is required when completely new memory tasks arise – such as entering a new environment or changing the animal’s goals – so that the animal is required to remember both the current task and the past ones separately, as opposed to slightly modifying the old task. In the context of our manifold picture, partial and rate remapping involve the deformation/refinement of the neural manifolds we are considering, while global remapping involves switching from one manifold to another.  Our results about the capacity for storing context follow from estimating the number of such manifolds that can be packed into the activity space in the presence of noise.  We can account for partial and rate remapping also in our framework by giving each manifold an additional width due to variations in the encoded structures that are not due to neural noise, but rather due to variations that arise from partial remapping. Thus, the qualitative structure of our results will remain unchanged by including the effects of partial remapping. We will discuss this extension in the revised submission – thank you for encouraging us to do so.  Note that, mechanistically, in the context of a network implementation, partial/rate remapping might involve the alteration of some continuous attractor implemented by the hippocampus, while global remapping would involve jumping from one continuous attractor to another.

---

### Decision · Program_Chairs · 2024-09-25

**Decision:**

Accept (oral)

**Comment:**

This paper develops a theory for understanding the tradeoffs between encoding space and encoding context in the hippocampus. The novel contributions include a characterization of how tuning width contributes to these functions and the trade-off between them. The authors use this theory to explain a number of experimentally-reported but not-well-theoretically-understood facts, including the functional role played by the gradient of tuning widths along the dorsal-ventral axis of the hippocampus, and why place cells might cluster near boundaries. The results and the theoretical concepts are well explained, broadly accessible and potentially impactful for computational and empirical neuroscience.